# When Do Neural Networks Learn World Models?

**Tianren Zhang** [1]  **Guanyu Chen** [1]  **Feng Chen** [1]

## Abstract

Humans develop *world models* that capture the underlying generation process of data. Whether neural networks can learn similar world models remains an open problem. In this work, we present the first theoretical results for this problem, showing that in a *multi-task* setting, models with a *low-degree bias* provably recover latent data-generating variables under mild assumptions– even if proxy tasks involve complex, non-linear functions of the latents. However, such recovery is sensitive to model architecture. Our analysis leverages Boolean models of task solutions via the Fourier-Walsh transform and introduces new techniques for analyzing invertible Boolean transforms, which may be of independent interest. We illustrate the algorithmic implications of our results and connect them to related research areas, including self-supervised learning, out-of-distribution generalization, and the linear representation hypothesis in large language models.

## 1. Introduction

Humans develop internal models of the world, extracting core concepts that generate perceptual data (Ha & Schmidhuber, 2018). Can neural networks do the same? With recent advances in large language models (LLMs), this question has garnered increasing attention (Bender et al., 2021; Mitchell, 2023). Understanding if and how neural networks learn human-like world models is crucial for building AI systems that are robust, fair, and aligned with human values (Hendrycks et al., 2023).

Empirical findings on world model learning have been mixed. Some studies suggest that medium-sized neural networks (Mikolov et al., 2013) and LLMs (Li et al., 2023; Bricken et al., 2023; Gurnee & Tegmark, 2024) learn abstract and interpretable features, indicating a non-trivial representation of data generation. Others, however, report a marked decline in LLM performance on novel tasks (Wu et al., 2023; Berglund et al., 2024; Mirzadeh et al., 2024), implying a lack of genuine world representations that enable human-level generalization in out-of-distribution settings.

Despite ongoing research, the theoretical foundations of learning world models remain unclear. Notably, even the term "world model" lacks a precise definition. This gives rise to several fundamental questions: what does it mean for neural networks to learn world models? When and why can they do so? More fundamentally, what constitutes a bona fide world model?

The **goal** of this work is to address these problems by introducing a formal framework for world model learning and presenting the first theoretical results in this area. Following the spirit of prior work (Ha & Schmidhuber, 2018; Li et al., 2023; Gurnee & Tegmark, 2024), we first show that latent variable models (Everett, 2013) provide a natural scaffold for formulating world model learning. Specifically, learning world models can be framed as achieving a non-trivial recovery of latent data-generating variables. However, a core challenge in this formulation arises from a well-known negative result showing that recovering true latents is generally impossible due to a fundamental issue of *non-identifiability* (Hyvärinen & Pajunen, 1999). That is, multiple solutions can fit the observed variables equally well, making true latent variables non-identifiable from observed data alone.

At first glance, the non-identifiability of latent variables may suggest a pessimistic outlook on learning world models. However, existing results overlook an important point: solutions that equally fit the data are not necessarily equivalent as *functions*. Thus, algorithms with implicit bias in function space, such as those employed in deep learning (Kalimeris et al., 2019; Goyal & Bengio, 2020), would favor certain solutions over others. In particular, we focus on a bias towards *low-complexity* functions, a phenomenon widely observed in neural networks and believed to be a key factor in the success of deep learning (Pérez et al., 2019; Huh et al., 2024; Goldblum et al., 2024). Yet, due to the lack of a well-established complexity measure for continuous functions, formalizing such complexity bias and analyzing its impact remains a challenging problem on its own (see

[1]Department of Automation, Tsinghua University, Beijing, China. Correspondence to: Feng Chen <chenfeng@mail.tsinghua.edu.cn>.

Section A for related work).

In this work, we circumvent this challenge by leveraging a simple yet important fact: while real-world data and latent variables may be continuous, all variables processed by neural networks are ultimately encoded as bit strings due to finite precision of computers. [1] This allows us to model all variables as *Boolean* without loss of generality. While this may seem a subtle difference (since we only lose the information that goes beyond machine precision), as we demonstrate in later sections, it turns out to provide surprisingly powerful machinery for defining and analyzing the *complexity* of solutions via the Fourier-Walsh transform of Boolean functions (O'Donnell, 2014). Building on this foundation, we present, for the first time, a nuanced perspective on learning world models that reveals an interplay between the low-complexity bias, proxy tasks, and model architecture. Our **main contributions** are:

1. In Section 2, we lay down general definitions of learning world models and discuss its core challenge posed by the non-identifiability of latent data-generating variables. This provides a foundation for future work to formally reason about learning world models and offers theoretical rigor to the recent scientific debate on this topic (Bender et al., 2021; Mitchell, 2023).

2. In Section 3, we introduce our Boolean function formulation and the corresponding complexity measures based on a notion of *realization degree*, offering a tractable approach for analyzing the impact of low-complexity bias on world model learning.

3. In Section 4, we present the first theoretical results on learning world models in the context of training on proxy tasks using observed data. We identify two critical factors for world model learning: (i) a *multi-task* learning setting; (ii) the low-complexity bias, instantiated by a *low-degree bias* of the model and a *low-degree task distribution*. Together, these factors ensure the identifiability of latent data-generating variables. Moreover, we show the provable benefits of learning world models in an out-of-distribution generalization setting (Abbe et al., 2023) and study the impact of model architecture under a notion of *basis compatibility* (see Figure 1 for a graphical summary of our results). Technically, our analysis relies on analyzing the degree properties of Boolean functions composed with invertible transforms, which may be of independent interest.

4. In Section 5, we illustrate the algorithmic implications of our results on two representative tasks: polynomial extrapolation (Xu et al., 2021) and learning physical

---

[1]Note that this differs from neural network quantization (Nagel et al., 2021; Gholami et al., 2021).

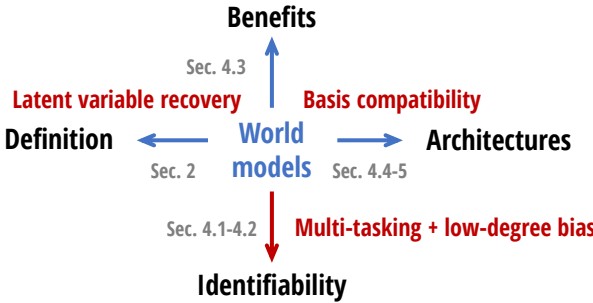

*Figure 1.* A graphical summary of our framework and main results.

laws (Kang et al., 2025). We show that architectures inspired by our analysis outperform conventional architectures such as ReLU MLPs and transformers (Vaswani et al., 2017) in these tasks.

## 2. Formulation of Learning World Models

How to define the world model and the problem of learning world models remains debatable to date. Yet, the term "world models" has been widely referred to in the literature as models that uncover the underlying *generation process* of data and maintain a representation of it (Ha & Schmidhuber, 2018; Gurnee & Tegmark, 2024; Richens & Everitt, 2024). For example, pioneering works by Li et al. (2023) and Nanda et al. (2023) define "world models" in board games as the board state that generates move sequences. This motivates a formulation of world model learning under the framework of latent variable models (Everett, 2013). To this end, we first define a general data generation process.

**Definition 2.1** (Data generation process). Let $\mathbf{x} \in \mathcal{X}$ be the observed data variables and let $\mathbf{z} \in \mathcal{Z}$ be the latent variables for some data space $\mathcal{X}$ and latent space $\mathcal{Z}$. The observed data are sampled as follows: (i) sample $\mathbf{z} \sim p(\mathbf{z})$ for some probability distribution $p$ over $\mathcal{Z}$; (ii) generate $\mathbf{x}$ through an invertible and non-linear function $\mathbf{x} = \psi(\mathbf{z})$.

This definition resembles the data generation process used in many latent variable models such as non-linear ICA (Hyvärinen & Pajunen, 1999), invariant feature learning (Arjovsky et al., 2019), and causal representation learning (Schölkopf et al., 2021). The main difference is that unlike these models, here we do not assume $p(\mathbf{z})$ to be any structured distribution.

Given Definition 2.1, a natural way to formalize "understanding" the generation process of data is to *invert* it, i.e., recover the latent variables $\mathbf{z}$ from the observed data $\mathbf{x}$. This leads to our basic formulation of learning world models.

**Definition 2.2** (Learning world models). Let $\mathcal{T}$ be a set of transforms $T : \mathcal{Z} \to \mathcal{Z}$. We say a representation $\Phi : \mathcal{X} \to \mathcal{Z}$ learns the world model up to $\mathcal{T}$ if there exists $T \in \mathcal{T}$ such

that $\Phi(\boldsymbol{x}) = T(\boldsymbol{z})$ for every $\boldsymbol{z} \in \mathrm{supp}(p)$ and $\boldsymbol{x} = \psi(\boldsymbol{z})$.

For instance, if $\mathcal{T}$ only contains the identity transform, then $\Phi(\mathbf{x})$ recovers $\mathbf{z}$ exactly. In general, we require $\mathcal{T}$ to contain only simple transform classes (e.g., linear transforms) for a meaningful recovery.

The main difficulty of this latent variable recovery problem is that $\mathbf{z}$ is, by definition, unobservable. To address this, we leverage the fact that models in practice are trained on some proxy tasks (e.g., next-token prediction) to learn representations of $\mathbf{x}$ implicitly. We formulate this as follows.

**Definition 2.3** (Task and realization). A *task* is defined as a function $h : \mathcal{X} \to \mathbb{R}$. For a task $h$, let

$$\mathcal{H}(h) := \{f : \mathcal{X} \to \mathbb{R} \mid f(\boldsymbol{x}) = h(\boldsymbol{x}), \\ \forall \boldsymbol{z} \in \mathrm{supp}(p), \boldsymbol{x} = \psi(\boldsymbol{z})\}. \quad (1)$$

If a function composition $f_{(1)} \circ \cdots \circ f_{(q)}$ is in $\mathcal{H}(h)$, then we say it is a *realization* of $h$.

**Flat and hierarchical realizations.** With definitions above, our main hope is that by training on some proper tasks $h$, the model can learn a *hierarchical realization* $g \circ \Phi \in \mathcal{H}(h)$ with a function $g : \mathcal{Z} \to \mathbb{R}$ and a representation $\Phi$ that learns the world model in the sense of Definition 2.2. However, a key challenge arises: by definition, all realizations have zero training error on $\mathrm{supp}(p)$ and are thus indistinguishable by their task performance. For example, every function $h^* \in \mathcal{H}(h)$ is itself a *flat realization* of $h$, i.e., without any explicit representation learning. Thus, by looking at the observable data alone, we have no reason to expect that $g \circ \Phi$ will be favored over $h^*$. Likewise, it is also unreasonable to expect that $g \circ \Phi$ should be favored over another hierarchical realization $g' \circ \Phi' \in \mathcal{H}(h)$ with $\Phi'$ not learning world models. Indeed, a well-known impossibility result in latent variable modeling shows that the true latent variables are *non-identifiable* when $\psi$ is a sufficiently flexible non-linear function of the latents (Hyvärinen & Pajunen, 1999; Khemakhem et al., 2020).

**Lemma 2.4** (Non-identifiability (Khemakhem et al., 2020)). *Let $\mathcal{Z} = \mathbb{R}^d$ and $\mathbf{z} \in \mathcal{Z}$ be a random vector of any distribution. Then, there exists an invertible transform $T : \mathcal{Z} \to \mathcal{Z}$ such that the components of $\mathbf{z}' = T(\mathbf{z})$ are independent, standard Gaussian variables.*

Lemma 2.4 indicates that we can construct new random variables $\Phi(\mathbf{x}) = \mathbf{z}'$ that have the same distribution $p$ as true latent variables $\psi^{-1}(\mathbf{x}) = \mathbf{z}$ (thus fitting the observed variables $\mathbf{x}$ equally well) by first transforming $\mathbf{z}$ to standard Gaussian variables, applying any orthogonal transform, and then inverting the transform (note that standard Gaussian distributions are invariant to orthogonal transforms). Applying this result to our context, we conclude that without additional assumptions, learning world models in the sense

of Definition 2.2 is not possible with a simple $\mathcal{T}$ by looking at the observed data $\mathbf{x}$ alone.

Given this result, one may naturally be pessimistic about the outlook on learning world models. However, recent studies on LLMs seem to suggest otherwise: instead of learning arbitrary non-linear representations of the observed data as Lemma 2.4 implies, LLMs turn out to often learn semantically meaningful, human-interpretable representations (Li et al., 2021; Bricken et al., 2023; Marks & Tegmark, 2023; Gurnee & Tegmark, 2024). This suggests that despite having a vast number of parameters and being capable of learning different task realizations (Reizinger et al., 2024), LLMs can still learn representations that are reasonably aligned with humans.

To address this puzzle, in this work we propose to incorporate the implicit bias of neural networks into analysis instead of using task performance as the only identifiability criterion. Specifically, we explore if the bias towards *low-complexity* realizations, a trait of both human reasoning and deep learning models, could be used to steer the learned realization towards non-trivial recovery of true latents. Yet, the main challenge is that for continuous functions with inputs in $\mathbb{R}^d$, we lack a well-established complexity measure that is amenable to analysis. For example, Kolmogorov complexity (Li et al., 2008) in algorithmic learning theory offers a unified framework for defining the complexity of any object, yet being uncomputable. Fortunately, in the following sections we will show that this problem could be circumvented by using Boolean functions.

## 3. Complexity Measures

Computers encode every object, including the observed data and variables learned by neural networks, in bit strings. Motivated by this fact, we could assume without loss of generality that both $\mathbf{x}$ and $\mathbf{z}$ in Definition 2.1 are Boolean (after some proper encoding). Formally, in the rest of the paper we will let $\mathcal{X} \subseteq \{-1, 1\}^m$ and $\mathcal{Z} = \{-1, 1\}^d$. Since $\psi$ is invertible, we have $m \geq d$. Unless otherwise mentioned, we will also assume $\mathrm{supp}(p) = \mathcal{Z}$ to ensure that all elements in $\mathcal{Z}$ can be sampled with non-zero probabilities.

A direct consequence of the Boolean modeling of variables is that all functions involved are *Boolean functions* (see Section B for a brief introduction). Notably, this itself does not resolve the non-identifiability issue stated in Section 2, as functions with Boolean inputs and outputs can still exhibit arbitrary nonlinearity. Instead, the primary advantage of this treatment is the useful machinery it provides for defining functional complexity, offered by the Fourier-Walsh transform of Boolean functions:

**Definition 3.1** (Fourier-Walsh transform (O'Donnell, 2014)). Every function $f : \{\pm 1\}^n \to \mathbb{R}$ can be uniquely

expressed as a multilinear polynomial

$$f(\boldsymbol{x}) = \sum\nolimits_{S \subseteq [n]} \hat{f}(S) \chi_S(\boldsymbol{x}), \qquad (2)$$

where $\boldsymbol{x} = (x_1, \ldots, x_n)$, $\chi_S(\boldsymbol{x}) = \prod_{i \in S} x_i$ are *parity functions*, and $\hat{f}(S) \in \mathbb{R}$ are the coefficients.

The Fourier-Walsh transform shows that every Boolean function can be represented as a *linear* combination of parity functions $\chi_S$ that capture all non-linear relationship between inputs and outputs. It can also be shown that parity functions are a *basis* of the vector space $\mathcal{F}^n := \{f : \{\pm 1\}^n \to \mathbb{R}\}$ of $n$-dimensional Boolean functions (see Section B for more details). By contrast, the space of arbitrary continuous functions does not have a similar, theory-friendly basis. See Section C for more discussion on the use of Boolean functions and complexity measures.

Given the Fourier-Walsh transform of a function, a natural measure of its complexity is its *degree*:

**Definition 3.2** (Degree). For every function $f : \{\pm 1\}^n \to \mathbb{R}$, its *degree* is defined by

$$\deg(f) = \max\{|S| : \hat{f}(S) \neq 0\}. \qquad (3)$$

For a Boolean function with multiple output dimensions, we define its degree by the sum of degrees of the Boolean functions mapping the input to each output coordinate.

*Remark* 3.3. Note that the degree of $f$ equals to the maximum degree of the basis functions it uses: $\deg(f) = \max\{\deg(\chi_S) : \hat{f}(S) \neq 0\}$. This interpretation will be useful when we consider different basis functions as in Section 4.4.

Intuitively, the degree of a Boolean function measures how non-linear it is. It can also be viewed as an approximation of Kolmogorov complexity if we treat parity functions $\chi_S$ as "function codes" with length $|S|$. Prior work has shown that many classes of neural networks indeed have a bias towards low-degree solutions for Boolean inputs (Abbe et al., 2023; Bhattamishra et al., 2023); in comparison, here we study whether a relevant low-complexity bias could implicitly lead to world model learning.

**Complexity measures for realizations.** Based on the notion of degree, we next introduce complexity measures to quantify the complexity of realizations. First, note that while degree can characterize function complexity, it fails to distinguish between different realizations, as they all behave identically when considered as a whole function. To overcome this limitation, we introduce *realization degree*.

**Definition 3.4** (Realization degree). Let $h$ be a task. For a realization $f_{(1)} \circ \cdots \circ f_{(q)}$ of $h$, its *realization degree* is defined as

$$\widehat{\deg}(f_{(1)} \circ \cdots \circ f_{(q)}) = \sum_{i \in [q]} \deg(f_{(i)}). \qquad (4)$$

For example, the realization degree of a flat realization $h^* \in \mathcal{H}(h)$ coincides with its degree; the realization degree of a hierarchical realization $h = g \circ \Phi$ is $\widehat{\deg}(g \circ \Phi) = \deg(g) + \deg(\Phi)$. Compared to degree, realization degree takes into account the cost of implementing each function in hierarchical realizations, which better captures representation learning in practice. Throughout this work, we say a model exhibits a **low-degree bias** if it minimizes the realization degree.

Finally, we introduce two definitions that are useful in characterizing the impact of the low-degree bias on flat and hierarchical realizations.

**Definition 3.5** (Min-degree solutions). For a task $h$, we define its *min-degree solutions* $\mathcal{H}_{\min}(h)$ as the set of functions in $\mathcal{H}(h)$ and that minimize the degree. We denote their degree by $\deg(\mathcal{H}_{\min}(h))$.

**Definition 3.6** (Conditional degree). For a task $h$ and a representation $\Phi : \mathcal{X} \to \mathcal{Z}$, let $h^* \in \mathcal{H}_{\min}(h)$ and assume that the set $\{g : \mathcal{Z} \to \mathbb{R} \mid g \circ \Phi \in \mathcal{H}(h)\}$ is not empty. The *conditional degree* of the task $h$ on the representation $\Phi$ is then defined as

$$\deg(h \mid \Phi) = \deg(h^*) - \max\{\deg(g) : g \circ \Phi \in \mathcal{H}(h)\}. \qquad (5)$$

By Definition 3.5, flat realizations with low-degree bias satisfy $h^* \in \mathcal{H}_{\min}(h)$. Analyzing the impact of representations is more involved; the main intuition behind Definition 3.6 is that a representation $\Phi$ only makes a task $h$ "simpler" if the conditional degree $\deg(h \mid \Phi) > 0$, i.e., solving the task on top of $\Phi$ yields a smaller degree compared to low-degree flat realizations. This notion will be explored further in the next section.

## 4. Theoretical Analysis

This section presents our main theoretical results. We first study how the low-degree bias introduced in Section 3 drives representation learning (Section 4.1). We then present our main results on learning world models, showing sufficient conditions for identifying latent data-generating variables (Section 4.2). Next, we show provable benefits of learning world models in an out-of-distribution generalization setting (Section 4.3). We conclude this section by a preliminary analysis on how the model architecture impacts world model learning (Section 4.4). All proofs are deferred to Section E. Finally, Section D presents numerical experiments that substantiate our theoretical results.

### 4.1. Low-Degree Bias Drives Representation Learning

As a warm-up, we first consider a basic question: why should neural networks, such as LLMs, learn *any* representation of data-generating variables when trained on proxy

tasks? Indeed, modern neural networks can often memorize the entire dataset (Zhang et al., 2017) or rely on superficial statistical patterns to solve tasks (Geirhos et al., 2019). This raises concerns about whether they truly understand the data despite generating plausible outputs (Bender et al., 2021).

Formally, this question can be modeled as a competition between flat realizations $h^* \in \mathcal{H}_{\min}(h)$ and hierarchical realizations $g \circ \Phi \in \mathcal{H}(h)$ in our formulation. Specifically, we seek to determine which realization minimizes the realization degree and is thus favored by the low-degree bias. Our first result shows that for any *single task* $h$, the flat realization is preferred.

**Theorem 4.1** (Single-task learning)**.** *Let $h$ be a task. Then, for every $h^* \in \mathcal{H}_{\min}(h)$, representation $\Phi : \mathcal{X} \to \mathcal{Z}$, and $g : \mathcal{Z} \to \mathbb{R}$ such that $g \circ \Phi \in \mathcal{H}(h)$, the following holds:*

$$\widehat{\deg}(h^*) \leq \widehat{\deg}(g \circ \Phi). \tag{6}$$

*Remark* 4.2. Intuitively, this result is a consequence of learning "redundant" representations: for any $h$, it suffices to learn every parity function $\chi_S$ with $\hat{h}(S) \neq 0$ in its Fourier-Walsh transform. Even if there is a good universal data representation, explicitly learning it is often not the best choice since it may involve irrelevant parity functions with $\hat{h}(S) = 0$, resulting in a larger realization degree.

However, the situation changes in the *multi-task* setting. Suppose there are $n$ distinct tasks $h_1, \ldots, h_n$. A flat realization solves each task independently by learning solutions $h_i^* \in \mathcal{H}_{\min}(h_i)$ for each $i \in [n]$, whereas a hierarchical realization can leverage the shared representation $\Phi$ across tasks, requiring only task-specific functions $g_i$ for $i \in [n]$. Our next theorem shows that in this setting, the hierarchical realization is favored if a sufficient number of tasks have a positive conditional degree.

**Theorem 4.3** (Multi-task learning)**.** *Let $h_1, \ldots, h_n$ be $n$ distinct tasks and let $h_i^* \in \mathcal{H}_{\min}(h_i), \forall i \in [n]$. Let $\Phi : \mathcal{X} \to \mathcal{Z}$ and $g_1, \ldots, g_n$ satisfy that $g_i \circ \Phi$ is an realization of $h_i, \forall i \in [n]$. Then, the following holds for every $h^* = (h_1^*, \ldots, h_n^*)$, $g = (g_1, \ldots, g_n)$, and $\Phi^* \in \mathcal{H}_{\min}(\Phi)$:*

$$\widehat{\deg}(h^*) - \widehat{\deg}(g \circ \Phi^*) \geq \sum_{i \in [n]} \deg(h_i \mid \Phi^*) - d^2. \tag{7}$$

Thus, if a sufficient number of tasks satisfy $\deg(h_i \mid \Phi) > 0$ and $\sum_{i \in [n]} \deg(h_i \mid \Phi) > d^2$, then $\widehat{\deg}(h^*) > \widehat{\deg}(g \circ \Phi)$, contrary to Theorem 4.1. We discuss two implications of these results:

- The contrast between the single-task and multi-task settings justifies the importance of multi-tasking in learning general-purpose representations, which has been conjectured by prior work (Radford et al., 2019; Huh et al.,

2024). Indeed, modern pre-training objectives such as next-token prediction and contrastive learning can be interpreted as solving a large number of prediction tasks simultaneously (Radford et al., 2019; Arora et al., 2019; Brown et al., 2020).

- Theorem 4.3 suggests that to facilitate the learning of a representation $\Phi$, proxy tasks should be chosen such that conditioning on $\Phi$ makes them less "complex". This provides a framework to reason about whether certain objectives in self-supervised learning (Liu et al., 2021) induce better representations than others. For instance, input reconstruction is often suboptimal as it permits a low-degree solution $h^*(\boldsymbol{x}) = \boldsymbol{x}$ without requiring any representation learning; masked image modeling (He et al., 2022) is likely more effective, as a representation that captures image semantics could significantly reduce solution complexity by filtering out pixel-to-pixel details.

### 4.2. Conditions for Learning World Models

We now move on to investigate whether the low-degree bias facilitates world model learning. While Theorem 4.3 shows that training on properly defined proxy tasks drives representation learning in the presence of the low-degree bias, it does not specify *which* representation is ultimately learned. In this section, we further explore the multi-task setting to address the key question: can we construct proxy tasks $h_1, \ldots, h_n$ to induce a representation $\Phi$ that learns the world models in the sense of Definition 2.2?

We begin by characterizing the space of all possible tasks. Given that the observed data is generated by $\mathbf{x} = \psi(\mathbf{z})$ with an invertible $\psi$, every task $h : \mathcal{X} \to \mathbb{R}$ can be equivalently defined via a latent-space function $h' : \mathcal{Z} \to \mathbb{R}$ as $h = h' \circ \psi^{-1}$. The space of all possible tasks can thus be represented as

$$\mathcal{F}^d \circ \psi^{-1} = \left\{ h' \circ \psi^{-1} \mid h' \in \mathcal{F}^d \right\}, \tag{8}$$

where $\mathcal{F}^d := \{h : \{\pm 1\}^d \to \mathbb{R}\}$ denotes the set of Boolean functions on $\mathcal{Z} = \{\pm 1\}^d$. Our next theorem shows that if proxy tasks are constructed by uniformly sampling from $\mathcal{F}^d \circ \psi^{-1}$, then as $n \to \infty$, all viable representations yield the same task-averaged realization complexity.

**Theorem 4.4** (Representational no free lunch)**.** *Let $h_1, \ldots, h_n$ be $n$ tasks that are independently and uniformly sampled from $\mathcal{F}^d \circ \psi^{-1}$. Then as $n \to \infty$, for any two representations $\Phi, \Phi'$ satisfying that there exists a bijective transform $T : \mathcal{Z} \to \mathcal{Z}$ such that $\Phi(\boldsymbol{x}) = T(\boldsymbol{z})$ for every $\boldsymbol{z} \in \mathrm{supp}(p)$ and $\boldsymbol{x} = \psi(\boldsymbol{z})$, the following holds:*

$$\lim_{n \to \infty} \frac{1}{n} \left( \widehat{\deg}(g \circ \Phi) - \widehat{\deg}(g' \circ \Phi') \right) = 0, \tag{9}$$

*where $g, g' \in (\mathcal{F}^d)^n$ satisfy that $g_i \circ \Phi$ and $g_i' \circ \Phi'$ are both realizations of $h_i$ for every $i \in [n]$.*

*Remark* 4.5. Note that the condition of the existence of a bijective transform $T$ is a minimal requirement for $\Phi(\mathbf{x})$ containing enough information for solving all tasks. The fact that all such representations have the same realization complexity suggests that the representation $\Phi$ induced by uniformly sampling from $\mathcal{F}^d \circ \psi^{-1}$ only learns the world model up to arbitrary bijective transforms.

Theorem 4.4 can be viewed as a "no free lunch"-like theorem for representation learning. The original no free lunch theorem (Wolpert, 1996) states that every learner's performance is equally good when averaged over a uniform distribution on learning problems; here we show that every viable representation is equally complex when averaged over a uniform distribution on tasks. As we will show in Section E.4, the technical intuition of this result is that every representation renders some tasks in the task space "simple" and others "complex", with the overall task-averaged complexity independent of the particular choice of the representation.

To overcome this result, we then move on to the *non-uniform* case where proxy tasks are still drawn from $\mathcal{F}^d$, but with different weights assigned to different functions. This setting is of more practical interest: prior work has reported various evidence suggesting that realistic tasks are often much more structured than being purely random (Whitley & Watson, 2005; Zhang et al., 2017). In particular, real-world data tend to be highly compressible, implying that low-complexity input-output maps occur more frequently than high-complexity ones (Dingle et al., 2018; Zhou et al., 2019; Goldblum et al., 2024). To formalize this, we define *k-degree tasks*.

**Definition 4.6** (*k*-degree tasks). Let $\mathcal{F}_k^d := \{h : \{\pm 1\}^d \to \mathbb{R} \mid \deg(h) \leq k\}$. We say a task $h$ is a *k*-degree task if $h \in \mathcal{F}_k^d \circ \psi^{-1} = \{h' \circ \psi^{-1} \mid h' \in \mathcal{F}_k^d\}$.

In other words, *k*-degree tasks can be solved by a function with degree not greater than $k$ on top of true latents. One can easily verify that $\mathcal{F}_k^d \subseteq \mathcal{F}_{k+1}^d$ for every $k \in [d-1]$ and $\mathcal{F}_d^d = \mathcal{F}^d$. Thus, uniform sampling from all possible tasks amounts to uniform sampling from $\mathcal{F}_d^d \circ \psi^{-1}$. *k*-degree tasks are also related to tasks with positive conditional degree, as shown by the following corollary.

**Corollary 4.7.** *For every task $h$ satisfying $\deg(h \mid \psi^{-1}) > 0$, we have $h \in \mathcal{F}_{d-1}^d \circ \psi^{-1}$.*

To capture the low-complexity bias in task sampling, we assign more weights to $\mathcal{F}_k^d \circ \psi^{-1} (k < d)$. Perhaps surprisingly, our next theorem shows that even a slight such preference on low-complexity tasks in the task distribution can induce world model learning up to simple transforms.

**Theorem 4.8** (World model learning). *Let $p_1, \ldots, p_d \in (0, 1)$ such that $\sum_{i \in [d]} p_i = 1$. Let $h_1, \ldots, h_n$ be $n$ tasks that are independently sampled as follows: (i) sample a degree $k \in [d]$ according to probabilities $\mathbf{Pr}[k = i] = p_i$;*

*(ii) uniformly sample a $k$-degree task. Let $(\Phi^*, g^*)$ be the minimizer of the following optimization problem:*

$$\min_{\Phi:\mathcal{X}\to\mathcal{Z}, g\in\mathcal{F}^d} \frac{1}{n} \widehat{\deg}(g \circ \Phi) \tag{10}$$
$$\text{s.t.} \quad g_i \circ \Phi \in \mathcal{H}(h_i), \, \forall i \in [n].$$

*Then as $n \to \infty$, $\Phi^*$ learns the world model up to negations and permutations, i.e., there exists a permutation $i_1, \ldots, i_d$ of $1, \ldots, d$ such that $\Phi_j^*(\boldsymbol{x}) \in \{\pm z_{i_j}\}$ for every $j \in [d]$, with $\boldsymbol{z} = \psi^{-1}(\boldsymbol{x})$.*

We make several remarks on this result:

- The reason why $k$-degree tasks facilitate world model learning is that, in effect, they induce a task distribution in which lower-degree tasks on true latents are drawn with larger probabilities than in the uniform setting. This overcomes the result in Theorem 4.4 by breaking the degree balance between different representations when averaged over the task distribution: representations capable of solving these lower-degree tasks in "cheaper" ways would now be favored.

- A limitation of Theorem 4.8 is that we requires a non-zero probability of explicitly sampling degree-1 tasks, in which latent variables are also task outputs. However, we emphasize that this probability is exponentially small as $d$ becomes large, and we conjecture that it can be completely removed in many settings. See Section E.6 for more discussion.

- Technically, in our proof we introduce a degree analysis of $k$-degree Boolean functions composed with invertible transforms (Lemma E.10), which may be of independent interest in analyzing similar problems.

**Connection to the linear representation hypothesis.** A number of recent mechanistic interpretation studies show that LLMs often represent abstract, interpretable features as *directions* in their intermediate representation space (Nanda et al., 2023; Marks & Tegmark, 2023; Gurnee & Tegmark, 2024). Theorem 4.8 could be viewed as a provable, Boolean version of the emergence of such linear representations: permutations and negations are precisely all degree-1 Boolean functions, a natural counterpart of degree-1 real polynomials (i.e., linear functions) in the real domain. The main significance of this result is that even when proxy tasks involve complex, non-linear functions over true latent variables $\boldsymbol{z}$, we can still recover $\boldsymbol{z}$ up to very simple transforms despite the presence of such nonlinearity.

### 4.3. Benefits of Learning World Models

Up to now, we have presented sufficient conditions for learning world models. As a complement to these results, this

section demonstrates provable *benefits* of learning world models in the context of an out-of-distribution generalization setting introduced by Abbe et al. (2023).

**Theorem 4.9** (Benefits of learning world models)**.** *Let the latent variables during training be uniformly sampled from the Hamming ball $B_r := \{z \in \{\pm 1\}^d \mid \#_{-1}(z) \leq r\}$ with $r < d$, and let the latent variables in test be uniformly sampled from $\mathcal{Z}$. Let $h : \{\pm 1\}^m \to \{\pm 1\}$ be a downstream task such that $h \circ \psi$ is a parity function with $\deg(h) = q > k = \lceil \log_2 \sum_{i=0}^r \binom{d}{i} \rceil$. Then, if $\deg(h \mid \psi^{-1}) \geq q - r$ and $\deg(h \circ \psi) > k - q + r$, the following holds: (i) the test mean square error (MSE) of any $h^* \in \mathcal{H}_{\min}(h)$ is larger than 1; (ii) let $\Phi^*$ be a representation that learns the world model up to negations and permutations as in Theorem 4.8 and let $g^*$ be a function such that $g^* \circ \Phi^* \in \mathcal{H}(h)$, then the test MSE of $g^* \circ \Phi^*$ is 0.*

*Remark* 4.10. As also noted by Abbe et al. (2023), a practical scenario reflected by sampling from $B_r$ is *length generalization* (Anil et al., 2022; Press et al., 2022). Here we show that in this setting, a hierarchical realization with the world model is provably more generalizable than any flat realization, despite that both of them achieve zero i.i.d. test error.

It has been widely believed that learning world models leads to better generalization (Li et al., 2023; Richens & Everitt, 2024; Yildirim & Paul, 2024). In comparison, Theorem 4.9 indicates that such benefits typically manifest when the conditional degree $\deg(h \mid \psi^{-1})$ of the downstream task $h$ is large enough. Technically, this is because for tasks with small $\deg(h \mid \psi^{-1})$, solutions using world models still involve high-degree parity functions, whose learning is hampered by the restricted sampling from $B_r$. As a practical example, semantical representations of images can make it easier to answer questions about high-level concepts (tasks with large $\deg(h \mid \psi^{-1})$), yet may make it harder to predict the intensity of a certain pixel (tasks with small or negative $\deg(h \mid \psi^{-1})$). Together with Theorems 4.4 and 4.8, this result suggests that a *low-degree task distribution* is essential for both learning world models and exploiting its advantage.

### 4.4. Impact of Model Architecture

In the above analysis, we study the role of the low-degree bias with the assumption that the task solutions perfectly adhere to it in the function space. Yet, practical training of neural networks is often more involved than this abstraction. Although neural networks are known as universal function approximators (Hornik et al., 1989; Funahashi, 1989), prior work has shown that models with different architectures may represent the same function differently (Raghu et al., 2021). In particular, embedded nonlinearities such as activation functions can steer how functions are represented by neural networks (Xu et al., 2020; Ziyin et al., 2020).

Motivated by this, in the following we analyze how different choices of *basis* in the Boolean function space can impact world model learning. Informally, one may also view neural networks as implementing functional bases through layer-by-layer function composition, and different neural network architectures may induce different bases in the function space (Teney et al., 2024).

Notably, for a given input dimension $n$, the Fourier-Walsh transform uses parity functions $\chi_S$ as a basis of the $2^n$-dimensional vector space $\mathcal{F}^n = \{f : \{\pm 1\} \to \mathbb{R}\}$ (cf. Definition 3.1). To obtain a different basis, we define a *basis transform* $U$, i.e., an invertible linear transform on $\mathcal{F}^n$ such that $\{U(\chi_S) \mid S \subseteq [n]\}$ is a basis of $\mathcal{F}^n$. The degree of a function $f : \{\pm 1\}^n \to \mathbb{R}$ under the new basis is then given by

$$\deg_U(f) := \max\{\deg(U^{-1}(\chi_S)) : \hat{f}(S) \neq 0\}, \quad (11)$$

where $U^{-1}$ reflects the cost of using the new basis to represent the original one. As a result, the low-degree bias under $\deg_U(\cdot)$ may deviate from that under $\deg(\cdot)$. To capture the effect of this, we introduce the notion of *basis compatibility*.

**Definition 4.11** (Compatibility)**.** We say a basis transform $U$ is *compatible* if $\deg(U(\chi_S)) = \deg(\chi_S)$, $\forall S \subseteq [n], \forall n$.

In other words, a basis transform is compatible if it preserves the degrees of all basis functions. Our next result shows the impact of basis compatibility on world model learning.

**Theorem 4.12** (Impact of basis compatibility)**.** *Consider the same setting as in Theorem 4.8 with $n \to \infty$. Let $U$ be a basis transform and let the degrees of $\Phi$ and $g$ be measured under the new basis $\{U(\chi_S)\}$. Then, (i) if $U$ is compatible, $\Phi^*$ learns the world model up to negations and permutations; (ii) there exists incompatible $U$ such that $\Phi^* = T \circ \psi^{-1}$, where $T$ is an invertible transform on $\mathcal{Z}$ satisfying $\max_{i \in [d]} \deg(T_i^{-1}) \geq k$.*

Theorem 4.12 implies that to facilitate world model learning, the model architecture should induce a basis that preserves the degree of the "natural" basis under which low-degree proxy tasks are drawn. Thus, we can also interpret basis compatibility as the compatibility between the model and the tasks. This explains why neural networks with different activation functions can exhibit different complexity biases (Abbe et al., 2023; Teney et al., 2024). Basis compatibility is also related to *algorithmic alignment* (Xu et al., 2020), which suggests that tasks with algorithmic structures aligned with the computational structures of neural networks could be learned more sample-efficiently. In comparison, we focus on the identifiability of world models instead of sample efficiency, and we provide a unified framework for analyzing similar concepts via the lens of the low-degree bias.

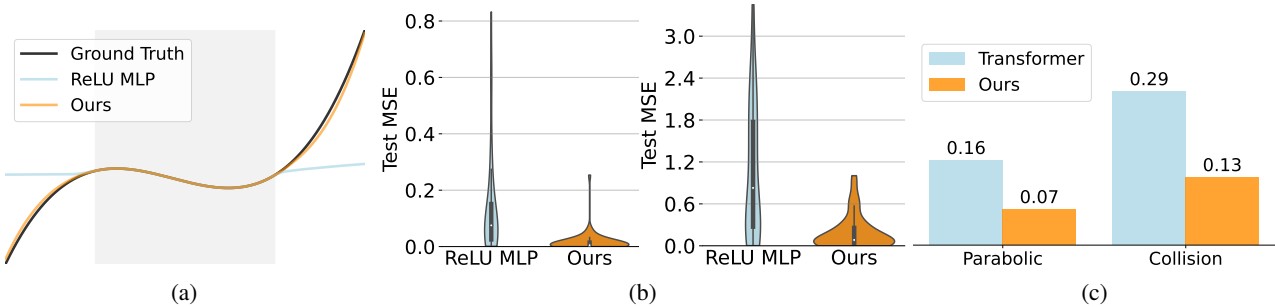

*Figure 2.* **Empirical results.** (a) An example of extrapolating a degree-3 polynomial. Shaded region indicates the training region. (b) Violin plots of the test mean square error (MSE) of ReLU MLPs and our models in extrapolating degree-2 (left) and degree-3 (right) polynomials. (c) Results for learning physical laws. Each column indicates the task and *out-of-distribution* test MSE averaged over 5 runs.

## 5. Algorithmic Implications

In this section, we illustrate the algorithmic implications of *basis compatibility* through two representative tasks: polynomial extrapolation (Xu et al., 2021) and learning physical laws (Kang et al., 2025; Motamed et al., 2025). The first task provides an illustrative example of our results in Section 4.4, while the second is of greater practical interest for learning real-world models and is related to recent efforts in developing video prediction models as world simulators (Brooks et al., 2024). While our experiments are currently limited in scale and mainly serve as proof-of-concept demonstrations, we view extending the ideas discussed in this section to broader contexts as a promising future direction.

### 5.1. Polynomial Extrapolation

We begin by a synthetic task in which we train multi-layer perceptrons (MLPs) to fit real polynomials $P_n(x) = \sum_{i=0}^{n} a_i x^n$ (see Section F.1 for details). The space of polynomials has a standard basis $\{1, x, x^2, \ldots\}$, making it a natural counterpart of the Boolean function space with the parity basis. Recovering all basis functions used by $P_n$ could generate any polynomial by linearly combining these basis functions, which enables extrapolation.

As shown by prior work (Xu et al., 2021) (see also Figure 2a), common ReLU MLPs cannot extrapolate degree-$k$ polynomials with $k > 1$ beyond the training region. Our framework explains why this happens via the incompatibility between ReLU and the polynomial basis: ReLU MLPs can learn a basis function $\hat{f}$ that approximates $f(x) = x^k$ arbitrarily well in any finite training region, but the *simplest* $\hat{f}$ that fits the data is not $f$ itself, which is expensive to represent using the composition of ReLU. As a result, the actually learned $\hat{f}$ differs from $f$ and hence does not extrapolate well.

Motivated by this explanation, here we provide a simple fix: replacing a portion of the ReLU activation functions with functions that are more compatible with the polynomial basis. In practice, we replace half of ReLU functions in each MLP layer by one of the two functions including the identity function $\sigma(x) = x$ and the quadratic function $\sigma(x) = x^2$. Representing $f(x) = x^k$ would be much easier using these functions, and we thus expect that a neural network with the low-complexity bias would then use them instead of the remaining ReLU despite the same expressive ability they have, resulting in $\hat{f} \approx f$. As shown in Figure 2b, this simple method indeed leads to significant improvement in extrapolation. Please see Section G.1 for more results and discussion.

One may wonder that in this task, we still need to know the basis of the task a priori to achieve basis compatibility. However, we emphasize that even without prior knowledge, we can still use different activation functions simultaneously and rely on the low-complexity bias of neural networks to *self-adaptively* select functions that are the most compatible with the task, as empirically shown in our experiments. We thus conjecture that this approach could be quite universal. Indeed, in the next section we will show that exactly the same approach can also bring benefits in a distinct scenario.

### 5.2. Learning Physical Laws

We now show that the approach in Section 5.1 also benefits a sequence prediction task aiming at learning physical laws. Correctly abstracting fundamental physical laws is essential for any model to be a real "world model" for the physical world (Motamed et al., 2025). Notably, a model that "understands" the physical laws is expected to generalize these laws to *unseen distributions* rather than fit them only in the training domain. Following Kang et al. (2025), we generate two types of object motion sequences reflecting basic physical laws: (i) single-object parabolic motion, and (ii) two-object elastic collision motion (see Section F.2 for more details). We train a transformer (Vaswani et al., 2017) to predict the motion of objects conditional on the first few frames in the motion sequence. The model is then evaluated in an *out-of-distribution generalization* setup with objects

having different initial velocities and sizes. To apply our method, we simply replace every MLP in the transformer with our modified MLP in Section 5.1 (we also replace the remaining ReLUs with GELUs).

As shown in Figure 2c, our model achieves lower prediction error than transformer in both settings. Note that both models achieve near zero training error. Thus, the fact that our model generalizes better out-of-distribution is not because it fits the data better, but due to it better capturing the underlying laws in object movements. See Section G.2 for the predicted motions from both models.

## 6. Limitations and Future Work

This work is an initial step towards formally understanding world model learning and the internal representations learned by neural networks, and we can see many exciting future directions. (i) Neural networks in practice learn *hierarchical representations* (Bengio et al., 2013), which may require a more structured modeling of the data generation model and the learned representations. (ii) We do not assume the latent variables to have any specific structure (e.g., causal relations between latent variables); extending our analysis to causal representation learning by combining our results and existing results in causal inference (Schölkopf et al., 2021; Richens & Everitt, 2024) is an interesting direction. (iii) We primarily focus on the low-complexity bias of neural networks; considering other implicit biases (Allen-Zhu & Li, 2023; Zhang et al., 2024) of neural networks and more fine-grained complexity measures are also promising directions for future work.

## Acknowledgements

This work was supported in part by the National Key Research and Development Program of China under STI 2030–Major Projects 2021ZD0200300, and in part by the National Key Research and Development Program of China No. 2024YDLN0006, and in part by the National Natural Science Foundation of China under Grant 62176133.

## Impact Statement

This paper presents work whose goal is to advance our understanding of the inner workings of deep neural networks. There are many potential societal consequences of our work, none which we feel must be specifically highlighted here.

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

# A. Related Work

**World models.** The term "world model" in the machine learning context originates from Ha & Schmidhuber (2018), who describe it as a human-like "mental model of the world" that learns an abstract representation of information flow and can be used to predict future events. This definition closely aligns with the concept of a "mental model" in cognitive science, i.e., an internal representation of external reality (Craik, 1967), making it naturally connected to the field of *representation learning* in machine learning (Bengio et al., 2013).

Recently, the remarkable capabilities of large language models (LLMs) have sparked a scientific debate on whether these models merely exploit superficial statistical patterns to generate predictions without genuine "understanding" of natural language (Bender et al., 2021; Mitchell, 2023), or whether they develop models that serve as compact and interpretable representations of the underlying data generation process. A series of studies have demonstrated the presence of internal representations in language models trained on synthetic tasks (Li et al., 2023; Nanda et al., 2023; Jin & Rinard, 2024). For real-world LLMs, research in mechanistic interpretability suggests that these models learn compact, interpretable, and causal features within their intermediate layers (Li et al., 2021; Bricken et al., 2023; Marks & Tegmark, 2023; Gurnee & Tegmark, 2024). At the same time, many studies report a significant decline in LLM performance on tasks that are assumed to be underrepresented in their pre-training distribution (Wu et al., 2023; Berglund et al., 2024; Mirzadeh et al., 2024).

Beyond sequence models, world models have also gained attention in reinforcement learning (Ha & Schmidhuber, 2018; LeCun, 2022; Xie et al., 2024), probabilistic learning (Friston et al., 2021; Wong et al., 2023), and causal discovery (Richens & Everitt, 2024). However, despite the growing number of empirical studies, the theoretical foundations of world model learning remain largely unexplored.

**Latent variable recovery.** Our definition of world models falls within a broad class of latent variable recovery problems (Everett, 2013), where observable data is generated by latent variables through an unknown generation function. It is well established that, without additional assumptions, recovering the true latent variables from observed data is generally impossible if the generation function is nonlinear (Hyvärinen & Pajunen, 1999; Khemakhem et al., 2020), a fundamental result in non-linear independent component analysis (non-linear ICA).

To overcome this impossibility, subsequent research has explored various structural assumptions on latent variables, such as conditional independence between the latent variables and an observable auxiliary variable (Hyvärinen et al., 2019; Khemakhem et al., 2020; Lee et al., 2021), distributional constraints (Zimmermann et al., 2021; Wei et al., 2021), and causal interventions (Von Kügelgen et al., 2021; Ahuja et al., 2023; von Kügelgen et al., 2023). Some studies have also linked these assumptions to contrastive learning (Hyvärinen et al., 2019; Tosh et al., 2021; Zimmermann et al., 2021). However, incorporating such structural assumptions often leads to complex and less scalable training paradigms compared to the pre-training framework of modern LLMs (i.e., next-token prediction). As a result, these studies do not directly address the central question of our work: *Can the ongoing paradigm of LLMs learn world models?* Meanwhile, the fact that LLMs already acquire non-trivial representations (Li et al., 2021; Bricken et al., 2023; Marks & Tegmark, 2023; Gurnee & Tegmark, 2024) suggests that they must leverage some form of *implicit bias* rather than explicit structural assumptions on input data, which motivates our study.

**Implicit bias of neural networks.** Overparameterized neural networks have been shown to possess the capacity to memorize entire training datasets (Zhang et al., 2017). However, their ability to generalize well in many settings suggests that they exhibit implicit preferences for certain solutions—commonly referred to as *implicit bias*. In simple models, such as linear models, random feature models, and two-layer neural networks, a body of theoretical work demonstrates that (stochastic) gradient descent imposes specific forms of implicit regularization on the learned solutions (Soudry et al., 2018; Gunasekar et al., 2018a;b; Bartlett et al., 2020; Chizat et al., 2020; Lyu et al., 2021; Allen-Zhu & Li, 2023; Andriushchenko et al., 2023; Abbe et al., 2023; Zhang et al., 2024).

Empirical studies further suggest that practical neural networks extend many of these implicit regularization effects through a form of *simplicity bias*, favoring "simpler" solutions over more complex ones (Pérez et al., 2019; Kalimeris et al., 2019; Xu et al., 2019; Bhattamishra et al., 2023; Huh et al., 2023; Zhao et al., 2024). However, the notion of "simplicity" is often defined empirically and varies across studies. In this work, we formalize the concept of simplicity within our theoretical framework and analyze its relationship to learning world models.

**Complexity measures.** Kolmogorov complexity (Li et al., 2008) in algorithmic learning theory provides a unified framework for quantifying the complexity of any object, including functions. However, it is not computable in general. In machine learning, conventional statistical learning theory typically employs the VC dimension as a complexity measure to bound the generalization error of models (Vapnik, 1999). More recent complexity measures include Rademacher and Gaussian complexities (Bartlett & Mendelson, 2002). However, these measures primarily assess the complexity of *function classes* rather than individual *functions*.

A growing body of research introduces complexity measures tailored for neural networks trained via (stochastic) gradient descent (Bartlett et al., 2017; Jacot et al., 2018; Zhou et al., 2019; Lotfi et al., 2022; Chatterjee & Sudijono, 2024), often leveraging them for generalization analysis. However, these measures inherently depend on the neural network parameterization, rather than capturing function complexity independently of specific parameterizations. Some studies propose alternative complexity metrics inspired by Kolmogorov complexity in the machine learning context (Xu et al., 2020; Liu et al., 2023), but these metrics are typically problem-specific and do not enable a direct complexity analysis in the function space, unlike the approach we take in this work. See Section C for more discussion on some commonly-used complexity measures.

## B. Preliminaries on Boolean Function Analysis

This section introduces basic definitions and properties of Boolean functions for readers who are not familiar with Boolean function analysis. For a more detailed introduction, we recommend the first few chapters of the book by O'Donnell (2014).

Following the convention in Boolean function analysis, throughout this work we use the term *Boolean functions* to refer to functions with the form

$$f : \{-1, 1\}^n \to \mathcal{Y}, \tag{12}$$

where $\mathcal{Y}$ could be any subspace of $\mathbb{R}^d$ for an arbitrary integer $d$, such as $\mathbb{R}$, $\{-1, 1\}^d$, etc.

There are different ways to represent the input *bits* in the above definiton. A natural way is to use $0$ and $1$ as elements of the field $\mathbb{F}_2$. In this way, a (single-output) Boolean function $f$ with $n$ input coordinates (bits) is represented by $f : \{0, 1\}^n \to \{0, 1\}$. It is also convenient to use $-1$ and $1$, thought as real numbers, and define $f$ as a function from $\{-1, 1\}^n$ to $\{-1, 1\}$. The latter representation can be easily transformed from the former one by the mapping $b \mapsto (-1)^b$ over $\{0, 1\}$. In our analysis, we will mostly use the latter representation as it is more compatible with the Fourier-Walsh transform of Boolean functions.

**Fourier-Walsh transform.** As in Definition 3.1, every function $\{-1, 1\}^n \to \mathbb{R}$ can be expressed as a multilinear polynomial, i.e., we have

$$f(\boldsymbol{x}) = \sum_{S \subseteq [n]} \hat{f}(S) \chi_S(\boldsymbol{x}), \tag{13}$$

where $\hat{f}(S)$ are Fourier-Walsh coefficients and $\chi_S(\boldsymbol{x}) = \prod_{i \in S} x_i$ are monomials, also called *parity functions*. The name parity function is due to the fact that it computes the logical parity or XOR of the bits. As an example, consider $f = \max_2$, i.e., the maximum function on 2 bits, and its Fourier-Walsh transform is $f(x_1, x_2) = \frac{1}{2} + \frac{1}{2} x_1 + \frac{1}{2} x_2 - \frac{1}{2} x_1 x_2$.

The existence of Fourier-Walsh transform can be proved by construction, using an interpolation method similar to constructing Lagrange polynomials. For each point $\boldsymbol{a} = (a_1, \ldots, a_n) \in \{-1, 1\}^n$, define

$$\mathbf{1}_{\boldsymbol{a}}(\boldsymbol{x}) = \prod_{i \in [n]} \frac{1 + a_i x_i}{2}. \tag{14}$$

It is easy to verify that $\mathbf{1}_{\boldsymbol{a}}(\boldsymbol{x})$ takes value $1$ when $\boldsymbol{x} = \boldsymbol{a}$ and $0$ otherwise. We then have

$$f(\boldsymbol{x}) = \sum_{\boldsymbol{a} \in \{-1, 1\}^n} f(\boldsymbol{a}) \mathbf{1}_{\boldsymbol{a}}(\boldsymbol{x}) \tag{15}$$

and arrive at a polynomial representation of $f$. Since any factors of $x_i^2, \forall i \in [n]$ can be replaced by $1$, we further know that this polynomial must be multilinear.

For functions $f : \mathbb{F}_2^n \to \mathbb{R}$, we can also define their Fourier-Walsh transforms by extending the $\chi_S$ notation using the mapping $b \mapsto (-1)^b$:

$$\chi_S(\boldsymbol{x}) = (-1)^{\sum_{i \in S} x_i}, \tag{16}$$

where $\boldsymbol{x} \in \mathbb{F}_2^n$. We can thus write the Fourier-Walsh transform of $f : \mathbb{F}_2^n \to \mathbb{R}$ in the same form as equation (13).

**Parity functions, orthogonality, and Parseval's Theorem.** We define the *inner product* $\langle \cdot, \cdot \rangle$ of functions $f : \{-1, 1\}^n \to \mathbb{R}$ and $g : \{-1, 1\}^n \to \mathbb{R}$ by

$$\langle f, g \rangle = \mathbb{E}_{\boldsymbol{x} \sim U(\{-1,1\}^n)}[f(\boldsymbol{x})g(\boldsymbol{x})]. \tag{17}$$

A key fact about parity functions is that they are orthogonal under the above definition: for every $S, S' \subseteq [n]$, we have

$$\langle \chi_S, \chi_{S'} \rangle = \begin{cases} 1, & S = S' \\ 0, & \text{otherwise} \end{cases}. \tag{18}$$

Consequently, if we consider the vector space $\mathcal{V}$ containing all functions $f : \{-1, 1\}^n \to \mathbb{R}$, then parity functions form an *orthonormal basis* of $\mathcal{V}$. It can be verified that the Fourier-Walsh coefficients in equation (13) satisfy

$$\hat{f}(S) = \langle f, \chi_S \rangle \tag{19}$$

for every $S \subseteq [n]$.

*Parseval's Theorem* shows that for every $f : \{-1, 1\}^n \to \mathbb{R}$,

$$\langle f, f \rangle = \sum_{S \subseteq [n]} \hat{f}(S)^2. \tag{20}$$

In particular, if $f$ is Boolean-valued, then $\langle f, f \rangle = 1$. The uniqueness of the Fourier-Walsh transform of Boolean functions can also be proved using the Parseval's Theorem (O'Donnell, 2014).

## C. Discussion on Complexity Measures

Besides the Boolean complexity measures considered by this work, there are also *continuous* approximations of Kolmogorov complexity, e.g., continuous complexity measures based on standard Fourier transform, the order of approximation polynomials, or compression (Xu et al., 2019; Pérez et al., 2019; Jiang et al., 2023; Teney et al., 2024). However, an important downside of these continuous complexity measures is that they do not apply to *arbitrary* continuous functions, neither constituting a *basis* of the continuous function space, which makes them less amenable to theoretical analysis.

For example, the linear function $f(x) = ax + b, x \in \mathbb{R}$ does not have a standard Fourier transform since it is not absolutely integrable. Even if we constrain it to some finite interval $[-L, L]$ and periodically extend it across the real axis, the resulting Fourier transform would involve frequencies up to infinity, suggesting infinite complexity if we use the highest frequency as the complexity measure. This is in contrast to the empirically observed simplicity bias of gradient descent, where linear models are often learned first (see e.g., Kalimeris et al. (2019)). The measure of polynomial order has a similar problem since not all continuous functions can be linearly represented by a set of polynomials. In fact, the degree of Boolean functions could be viewed as a "discrete" version of polynomial order, but it is more principled since the Boolean function space indeed has a polynomial basis given by the Fourier-Walsh transform. Finally, we found other compression-based approximations of Kolmogorov complexity hard to analyze theoretically since they rely on specific compression algorithms.

## D. Numerical Experiments

This section presents numerical experiments that corroborate our theoretical results. In all experiments, we train MLPs using stochastic gradient descent (SGD) and binary data from $\{\pm 1\}^m$, generated by the data generation model in Table 1 with $d = m = 10$. A summarization of results is as follows:

- **Low-degree bias of neural networks:** Figure 3 shows that MLPs indeed have a low-degree bias when solutions with different degrees exist for the same training distribution.

- **Impact of multi-task training on identifying data-generating variables:** Figure 4 shows that the identification error of data-generating variables significantly decreases as more training tasks are drawn, while single-task learning does not lead to similar identification.

- **Identifying data-generating variables benefits out-of-distribution (OOD) generalization:** Figure 5 shows that in a Hamming sampling setting as in Theorem 4.9, MLPs with a representation $\Phi$ that identifies true data-generating variables achieve superior OOD generalization compared to MLPs without such a representation.

Please refer to the figures for more details and discussion.

It is worth noting that in all experiments, we use MLPs with quadratic activation $\sigma(x) = x^2$ to ensure the compatibility between the model and the Fourier-Walsh transform (i.e., the *polynomial* expression) of Boolean functions. Our preliminary experiments showed that the low-degree bias and the identification error are very sensitive to the activation function used, which is consistent with our basis compatibility analysis in Section 4.4. This also explains the phenomenon reported by Abbe et al. (2023) that MLPs with ReLU activation do not exhibit a "pure" low-degree bias for Boolean inputs.

*Table 1.* The data generation model $\boldsymbol{x} = \psi(\boldsymbol{z})$ used in numerical experiments.

| $x_1$ | $x_2$ | $x_3$ | $x_4$ | $x_5$ | $x_6$ | $x_7$ | $x_8$ | $x_9$ | $x_{10}$ |
|---|---|---|---|---|---|---|---|---|---|
| $z_1$ | $z_1 z_2$ | $z_1 z_2 z_3$ | $z_1 z_2 z_3 z_4$ | $z_1 z_2 z_3 z_4 z_5$ | $z_2 z_3 z_4 z_5 z_6$ | $z_3 z_4 z_5 z_6 z_7$ | $z_4 z_5 z_6 z_7 z_8$ | $z_5 z_6 z_7 z_8 z_9$ | $z_6 z_7 z_8 z_9 z_{10}$ |

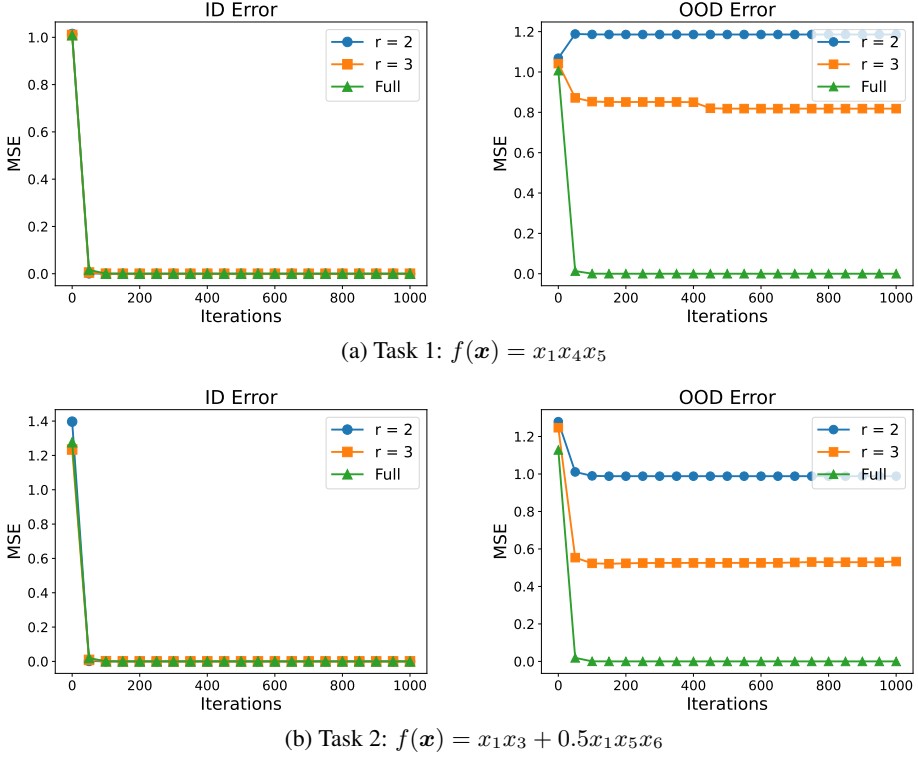

(a) Task 1: $f(\boldsymbol{x}) = x_1 x_4 x_5$

(b) Task 2: $f(\boldsymbol{x}) = x_1 x_3 + 0.5 x_1 x_5 x_6$

*Figure 3.* **Low-degree bias of neural networks:** Training latents are uniformly sampled from the Hamming ball $B_r = \{\boldsymbol{z} \in \{\pm 1\}^d \mid \#_{-1}(\boldsymbol{z}) \leq r\}$; the out-of-distribution (OOD) test latents are uniformly sampled from $\{\pm 1\}^d$ ($d = 10$). For small $r$, *low-degree realizations* of tasks (target functions) exist in the training distribution. A model that learns these low-degree realizations would have low in-distribution (ID) error yet high OOD error due to the loss of high-degree components in target functions. We observe that while SGD-trained MLPs learn ground-truth target functions when $r = 10$ (denoted by "Full" in the legends), they indeed fit low-degree realizations for training data with $r = 2$ and $r = 3$, resulting in high OOD error. We separately train a three-layer MLP for each task and for each $r$.

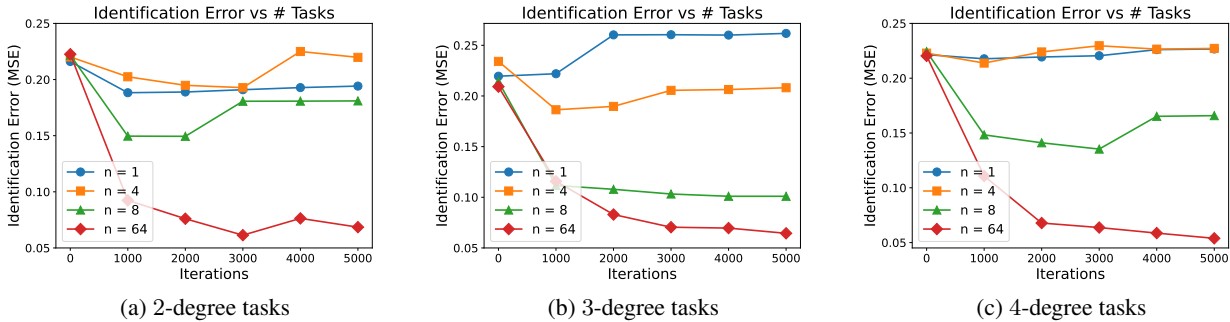

(a) 2-degree tasks        (b) 3-degree tasks        (c) 4-degree tasks

*Figure 4.* **Impact of multi-task training on identifying data-generating variables:** The identification error of data-generating variables *decreases* as the task number $n$ *increases*, measured by the MSE of the linear probe trained to fit true latent variables $\mathbf{z}$ given the learned representation $\Phi(\mathbf{x})$. Each task is a randomly sampled parity function of $\mathbf{z}$ with degree not larger than $k$ for $k$-degree tasks. Compared to single-task training ($n = 1$), multi-task training with $n = 64$ significantly reduces the identification error for all $k$-degree training task distributions with $k = 2, 3, 4$, corroborating *Theorem 4.3* and *Theorem 4.8* in the main text. Both the representation $\Phi$ and each task-specific function $g_i$, $i \in [n]$ are instantiated as a three-layer MLP.

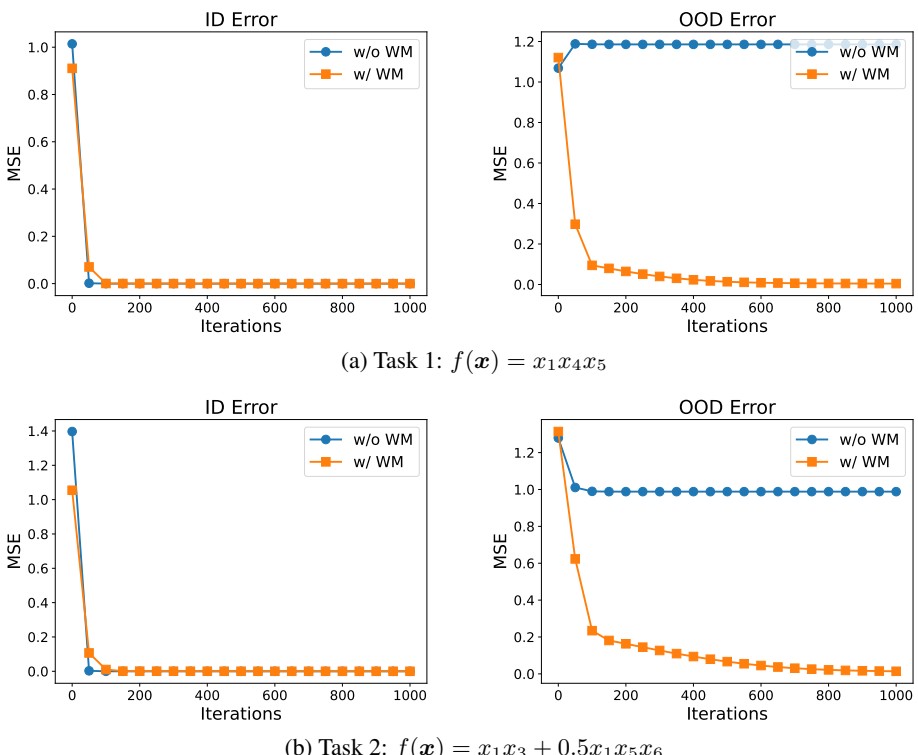

(a) Task 1: $f(\boldsymbol{x}) = x_1 x_4 x_5$

(b) Task 2: $f(\boldsymbol{x}) = x_1 x_3 + 0.5 x_1 x_5 x_6$

*Figure 5.* **Identifying data-generating variables benefits out-of-distribution (OOD) generalization:** For the same tasks considered in Figure 3 with $r = 2$, SGD-trained MLPs on top of an oracle representation that identifies true data-generating variables (denoted by "w/ WM" in the legends) achieve OOD generalization. By contrast, SGD-trained MLPs without the representation (denoted by "w/o WM" in the legends) fail. This corroborates *Theorem 4.9* in the main text. We separately train an MLP for each task and for the setting with/without the oracle representation.

# E. Proofs

This section provides complete proofs of all theorems in the main text, organized as follows.

- In Section E.1, we introduce some definitions and technical lemmas.

- In Section E.2, we provide the proof of Theorem 4.1.

- In Section E.3, we provide the proof of Theorem 4.3.

- In Section E.4, we provide the proof of Theorem 4.4.

- In Section E.5, we provide the proof of Corollary 4.7.

- In Section E.6, we provide the proof of Theorem 4.8.

- In Section E.7, we provide the proof of Theorem 4.9.

- In Section E.8, we provide the proof of Theorem 4.12.

**Notation and conventions.** We use $[n]$ to denote the set $\{1, \ldots, n\}$ for positive integers $n$. For a set $S$, we denote its cardinality by $|S|$. For a probability distribution $p$ over some set $S$, we denote by $\mathrm{supp}(p) := \{s \in S \mid p(s) > 0\}$ its support. For $n$ functions $f_1, \ldots, f_n$, we use $f = (f_1, \ldots, f_n)$ to denote the multi-output function satisfying that $f(\boldsymbol{x}) = (f_1(\boldsymbol{x}), \ldots, f_n(\boldsymbol{x}))$; conversely, for a function $f$ with $n$ output dimensions, we use $f_i$ to denote the function mapping the inputs of $f$ to its $i$-th output dimension for $i \in [n]$. As defined in the main text, we use the notation $\mathcal{F}^n = \{f : \{\pm 1\}^d \to \mathbb{R}\}$ and $\mathcal{F}_k^n = \{f : \{\pm 1\}^d \to \mathbb{R} \mid \deg(f) \leq k\}$ for positive integers $n$. Note that although both $\mathcal{F}^n$ and $\mathcal{F}_k^n$ are infinite, only finite functions in them are implementable by computers due to bounded precision. Therefore, in our proofs we will treat them as finite yet exponentially large sets. This enables us to use, e.g., $|\mathcal{F}^n|$ and $\sum_{h' \in \mathcal{F}^n} \deg(h')$ in our proofs and helps avoid non-essential technical nuances.

## E.1. Technical Lemmas

This section presents additional definitions and technical lemmas that may come in handy in our proofs. We begin by introducing a lemma that upper-bounds the degree of min-degree solutions for every task with $d$-dimensional latent variables.

**Lemma E.1.** *Suppose that the input data variable* $\mathbf{x} \in \mathcal{X} \subseteq \{-1, 1\}^m$ *is generated as in Definition 2.1 by a $d$-dimensional lantent variable* $\mathbf{z} \in \mathcal{Z} = \{-1, 1\}^d$, $d \leq m$. *Then, for every task* $h : \mathcal{X} \to \{-1, 1\}$, *we have*

$$\deg(\mathcal{H}_{\min}(h)) \leq d. \tag{21}$$

*Proof.* The case of $d = m$ is trivial, so in what follows we consider the case of $d > m$. We first prove the following lemma:

**Lemma E.2.** *For every* $S = \{i_1, \ldots, i_{d+1}\} \subseteq [m]$ *with* $|S| = d+1$, *there exist* $b_1, \ldots, b_{d+1}$ *with* $b_i \in \{-1, 1\}, \forall i \in [d+1]$ *such that* $\prod_{j \in [d+1]} (x_{i_j} + b_j) = 0$ *holds for every* $\boldsymbol{x} \in \mathcal{X}$.

*Proof of Lemma E.2.* We can prove Lemma E.2 by contradiction: assume that it does not hold, i.e., for some $S$, there exists $\boldsymbol{x} \in \mathcal{X}$ such that $\prod_{j \in [d+1]} (x_{i_j} + b_j) \neq 0$, then we must have $x_{i_j} = b_j, \forall j \in [d+1]$. Since $b_1, \ldots, b_{d+1}$ are arbitrary, by applying this argument to every $(b_1, \ldots, b_{d+1}) \in \{-1, 1\}^{d+1}$ we can find $2^{d+1}$ elements in $\mathcal{X}$ that differ in at least one coordinate in $S$, which yields $|\mathcal{X}| \geq 2^{d+1}$. On the other hand, recall that we assume $\mathrm{supp}(\mathbf{z}) = \mathcal{Z}$. Due to the invertibility of $f$, we have $|\mathcal{X}| = |\mathcal{Z}| = 2^d$, which contradicts $|\mathcal{X}| \geq 2^{d+1}$. Hence, the initial assumption is false and Lemma E.2 holds. $\square$

Applying Lemma E.2, we have that for every degree-$d + 1$ subset $S = \{i_1, \ldots, i_{d+1}\} \subseteq [m]$,

$$\prod_{j \in [d+1]} (x_{i_j} + b_j) = \prod_{j \in S} x_j + \sum_{S' \subset S, |S'| \leq d} b_{S'} \prod_{k \in S'} x_k = 0, \ \forall \boldsymbol{x} \in \mathcal{X} \tag{22}$$

holds for some $b_1, \ldots, b_{d+1}$, where $b_{S'} \in \{-1, 1\}$ for every $S' \subset S$. This means that we can thus replace every degree-$d + 1$ monomial $\chi_S(\boldsymbol{x}) = \prod_{j \in S} x_j$ by a degree-$d$ polynomial $-\sum_{S' \in 2^S, |S'| \leq d} b_{S'} \prod_{k \in S'} x_k$. By iteratively using this

replacement in the Fourier-Walsh transform of $h$ (Definition 3.1), one can eventually obtain a polynomial with degree $d$ or less without changing the value of the function on every $\boldsymbol{x} \in \mathcal{X}$. This completes the proof. $\qquad\square$

*Remark* E.3. Without the latent structure, a trivial upper bound of the min-degree solution of any task $h$ is $\deg(\mathcal{H}_{\min}(h)) \leq m$. Hence, Lemma E.1 is an example of how the min-degree bias can exploit the latent structure of data by favoring low-degree solutions with degree independent of the data dimension $m$.

Before presenting our next lemma, we first introduce the concept of influence for Boolean functions.

**Definition E.4** (Influence). Let $f : \{-1, 1\}^n \to \{-1, 1\}$ be a Boolean function. Then, the *influence* of coordinate $i, i \in [n]$ on $f$ is defined as

$$\mathrm{Inf}_i(f) = \mathbf{Pr}_{\boldsymbol{x} \sim U(\{-1,1\}^n)}[f(\boldsymbol{x}) \neq f(\boldsymbol{x}^{\oplus i})], \tag{23}$$

where $\boldsymbol{x}^{\oplus i} = (x_1, \ldots, x_{i-1}, -x_i, x_{i+1}, \ldots, x_n)$.

By Parseval's Theorem, we have a formula between influence and the Fourier-Walsh coefficients (O'Donnell, 2014).

**Lemma E.5.** *For $f : \{-1, 1\}^n \to \{-1, 1\}$ and every $i \in [n]$, the following holds:*

$$\mathrm{Inf}_i(f) = \sum_{S \in \{S' \subseteq [n] \,|\, i \in S'\}} \hat{f}(S)^2, \tag{24}$$

*where $\hat{f}(S)$ is the Fourier-Walsh coefficients of $f$ as in Definition 3.1.*

With the above definitions, our next lemma introduces a necessary and sufficient condition of bijective Boolean functions being degree-1, based on the restricted influence of all input coordinates.

**Lemma E.6.** *Let $f : \{-1, 1\}^n \to \{-1, 1\}^n$ be a bijective function and let $1 \leq k \leq n - 1$ be an integer. Then, we have $\deg(f_i) = 1, \forall i \in [n]$ if and only if for every $S \subset [n]$ with $|S| = k$, there exists $T \subset [n]$ with $|T| = k$ such that:*

- *for every $j \in T$, $\mathrm{Inf}_i(f_j) > 0$ for at least one $i \in S$;*

- *for every $j \in [n] \setminus T$, $\mathrm{Inf}_i(f_j) = 0$ for every $i \in S$.*

*Proof.* Note that $\deg(f_i) = 1, \forall i \in [n]$ together with the fact that $f$ is bijective indicates the existence of a permutation $i_1, \ldots, i_n$ of $1, \ldots, n$ such that $f_j(\boldsymbol{x}) = x_{i_j}$ or $f_j(\boldsymbol{x}) = -x_{i_j}$ for every $j \in [n]$, which trivially gives the result. In the following we prove the other direction. Note that the case of $k = 1$ is trivial. Hence, to prove that $\deg(f_i) = 1$, it suffices to prove (*): for every $i \in [n]$, there exists $j \in [n]$ such that $\mathrm{Inf}_i(f_j) > 0$ and $\mathrm{Inf}_i(f_m) = 0, \forall m \neq j, m \in [n]$, for every $2 \leq k \leq n - 1$.

We first prove that for every $S$, $T$ is unique, by contradiction. Suppose that for some $S$, $T$ is not unique, i.e., there exists $T' \neq T \subset [n]$ such that $T'$ satisfies the condition. Then, by Definition E.4, changing the values of the coordinates $x_i, i \in S$ can change only the values of $f_j(\boldsymbol{x}), j \in T \cap T'$ but not the values of other $f_j(\boldsymbol{x}), j \in [n] \setminus (T \cup T')$. This results in $|\{f(\boldsymbol{x}), \boldsymbol{x} \in \mathcal{X}\}| \leq 2^{n-|S|} \cdot 2^{|T \cap T'|} = 2^{n-k+|T \cap T'|} < 2^n$, contradicting the bijectivity of $f$. Therefore, the assumption is false and $T$ is unique. This allows us to define a mapping $\phi : S \mapsto T$.

Let $\mathcal{S}_k = \{S \subset [n] \mid |S| = k\}$ be the set of all subsets of $[n]$ with cardinality $k$. We then prove that $\phi : S \mapsto T$ is bijective on $\mathcal{S}_k$. To this end, it suffices to show that every $S \neq S' \in \mathcal{S}_k$ are mapped to different $T$. This can be similarly proved by contradiction as in proving the uniqueness of $T$: if it is false, then there exists a subset $S'' = S \cup S'$ and $T$ such that $\mathrm{Inf}_i(f_j) = 0, \forall i \in S'', j \in [n] \setminus T$. Then, by Definition E.4, changing the values of the coordinates $x_i, i \in S''$ can change only the values of $f_j(\boldsymbol{x}), j \in T$ but not the values of other $f_j(\boldsymbol{x}), j \notin T$. This results in $|\{f(\boldsymbol{x}), \boldsymbol{x} \in \mathcal{X}\}| \leq 2^{|T|} \cdot 2^{n-|S''|} = 2^{n+k-|S''|} < 2^n$, contradicting the bijectivity of $f$. Therefore, the assumption is false and every $S \neq S' \in \mathcal{S}_k$ are mapped to different $T$.

Given that $\phi : S \mapsto T$ is bijective, we know that for every subset $\mathcal{S} \subseteq \mathcal{S}_k$, there exists a unique subset $\mathcal{T} = \{T = \phi(S) \mid S \in \mathcal{S}\} \subseteq \mathcal{S}_k$ such that $|\mathcal{S}| = |\mathcal{T}|$. We then move on to prove the proposition (*) by contradiction: suppose it is false, i.e., for some $i \in [n]$, there exists $U \subset [n]$ with $|U| \geq 2$ such that $\mathrm{Inf}_i(f_j) > 0$ for every $j \in U$ and $\mathrm{Inf}_i(f_j) = 0$ for every $j \in [n] \setminus U$. If $|U| > k$, then for any $S$ such that $i \in S$, there does not exist a feasible $T$, which is a contradiction. If $2 \leq |U| \leq k$, consider $\mathcal{S} = \{S \subset \mathcal{S}_k \mid i \in S\}$ with $|\mathcal{S}| = C_{n-1}^{k-1}$. By the definition of $\phi$, we know that $\mathcal{T} \subseteq \{T \subset \mathcal{S}_k \mid U \subseteq T\}$, which gives $|\mathcal{T}| \leq C_{n-|U|}^{k-|U|} < C_{n-1}^{k-1} = |\mathcal{S}|$. Thus, the assumption is false and proposition (*) is true. This completes the proof. $\qquad\square$

Our next lemma shows that any bijective transform on $\{-1, 1\}^d$ can induce a bijective transform on $\mathcal{F}^d$.

**Lemma E.7.** *Let $T : \{-1, 1\}^d \to \{-1, 1\}^d$ be a bijective transform. Then, $\mathcal{F}^d \circ T = \mathcal{F}^d$.*

*Proof.* It suffices to show that the mapping $h' \mapsto h' \circ T$ for $h' \in \mathcal{F}^d$ is bijective. On one hand, it is obvious that $h' \circ T \in \mathcal{F}^d$ for every $h' \in \mathcal{F}^d$. On the other hand, for each $h' \in \mathcal{F}^d$, there exists $h'' = h' \circ T^{-1}$ such that $h'' \circ T = h'$. This completes the proof. $\square$

We then present a lemma from Abbe et al. (2023) that guarantees the uniqueness of low-degree solutions when the training data is sampled from a Hamming ball.

**Lemma E.8** (Abbe et al. (2023), Theorem 5.1). *Consider a Boolean function $f : \{\pm 1\}^d \to \mathbb{R}$. Then, there exists a unique function $f_r : \{\pm 1\}^d \to \mathbb{R}$ such that for every $z \in B_r := \{z \in \{\pm 1\}^d \mid \#_{-1}(z) \leq r\}$, we have $f_r(z) = f(z)$ and $\deg(f_r) \leq r$.*

Our next lemma shows that a bijection on $\mathcal{F}^n$ that preserves the degree of all parity functions preserves the degree of all functions.

**Lemma E.9.** *Let $U : \mathcal{F}^n \to \mathcal{F}^n$ be an invertible linear transform. If $\deg(U(\chi_S)) = \deg(\chi_S)$ for every $S \subseteq [n]$, then we have*

$$\deg(U(f)) = \deg(f), \ \forall f \in \mathcal{F}^n. \tag{25}$$

*Proof.* By the linearity of $U$, we have

$$U(f) = U\left(\sum_{S \subseteq [n]} \hat{f}(S)\chi_S\right) = \sum_{S \subseteq [n]} \hat{f}(S)U(\chi_S). \tag{26}$$

It then follows from Definition 3.2 that

$$\deg(U(f)) = \max\left\{\deg(U(\chi_S)) : \hat{f}(S) \neq 0\right\} = \max\left\{\deg(\chi_S) : \hat{f}(S) \neq 0\right\} = \deg(f). \tag{27}$$

This completes the proof. $\square$

### E.2. Proof of Theorem 4.1

*Proof.* Since $g \circ \Phi$ is an realization of $h$, we have $g \circ \Phi \in \mathcal{H}(h)$. This gives

$$\widehat{\deg}(h^*) = \deg(h^*) = \deg(\mathcal{H}_{\min}(h)) \leq \deg(g \circ \Phi). \tag{28}$$

Thus, it suffices to prove that $\deg(g \circ \Phi) \leq \widehat{\deg}(g \circ \Phi)$ for every $g : \mathcal{Z} \to \mathbb{R}$ and $\Phi : \mathcal{X} \to \mathcal{Z}$. Let the Fourier-Walsh transform of $g$ and $\Phi_i, i \in [d]$ be

$$g(z) = \sum_{G \subseteq [d]} \hat{g}(G) \prod_{i \in G} z_i \tag{29}$$

and

$$\Phi_i(x) = \sum_{S \subseteq [m]} \hat{\Phi}_i(S) \prod_{j \in S} x_j, \tag{30}$$

respectively. Plugging (30) into (29) with $z_i = \Phi_i(x)$ gives

$$(g \circ \Phi)(x) = \sum_{G \subseteq [d]} \hat{g}(G) \prod_{i \in G} \left(\sum_{S \subseteq [m]} \hat{\Phi}_i(S) \prod_{k \in S} x_k\right) \tag{31}$$

$$= \sum_{G \in \{G' \subseteq [d], \hat{g}(G') \neq 0\}} \hat{g}(G) \prod_{i \in G} \left(\sum_{S \in \{S' \subseteq [m], \hat{\Phi}_i(S') \neq 0\}} \hat{\Phi}_i(S) \prod_{k \in S} x_k\right). \tag{32}$$

We thus have

$$\deg(g \circ \Phi) \leq \max_{G \in \{G' \subseteq [d], \hat{g}(G') \neq 0\}} \sum_{i \in G} \max_{S \in \{S' \subseteq [m], \hat{\Phi}_i(S') \neq 0\}} |S| \tag{33}$$

$$\leq \max_{G \subseteq [d]} \sum_{i \in G} \max_{S \in \{S' \subseteq [m], \hat{\Phi}_i(S') \neq 0\}} |S| \tag{34}$$

$$\leq \sum_{i \in [d]} \max_{S \in \{S' \subseteq [m], \hat{\Phi}_i(S') \neq 0\}} |S| \tag{35}$$

$$= \sum_{i \in [d]} \deg(\Phi_i). \tag{36}$$

On the other hand, Definition 3.4 gives

$$\widehat{\deg}(g \circ \Phi) = \deg(g) + \deg(\Phi) = \deg(g) + \sum_{i \in [d]} \deg(\Phi_i) \tag{37}$$

$$\geq \sum_{i \in [d]} \deg(\Phi_i). \tag{38}$$

Therefore, we have $\deg(g \circ \Phi) \leq \widehat{\deg}(g \circ \Phi)$. This completes the proof. $\square$

### E.3. Proof of Theorem 4.3

*Proof.* By Lemma E.1, we can upper-bound the degree of each $\Phi_j^*, j \in [d]$ by

$$\deg(\Phi_j^*) \leq d. \tag{39}$$

Expanding the LHS of (7) then gives

$$\widehat{\deg}(h^*) - \widehat{\deg}(g \circ \Phi^*) = \sum_{i \in [n]} \deg(h_i^*) - \sum_{i \in [n]} \deg(g_i) - \sum_{j \in [d]} \deg(\Phi_j^*) \tag{40}$$

$$\geq \sum_{i \in [n]} \deg(h_i^*) - \sum_{i \in [n]} \deg(g_i) - d^2. \tag{41}$$

Note that $h_i^* \in \mathcal{H}_{\min}(h)$ and $g_i \circ \Phi$ is an realization of $h_i$ for every $i \in [n]$. By Definition 3.6, we have

$$\deg(h_i \mid \Phi^*) = \deg(h_i^*) - \deg(g_i), \forall i \in [n]. \tag{42}$$

Plugging equation (42) into (41) completes the proof. $\square$

### E.4. Proof of Theorem 4.4

*Proof.* Note that for every task $h \in \mathcal{F}^d \circ \psi^{-1}$, we can write $h = h' \circ \psi^{-1}$ for some $h' \in \mathcal{F}^d$. Thus, for every $g : \mathcal{Z} \to \mathbb{R}$ and $\Phi : \mathcal{X} \to \mathcal{Z}$ such that $g \circ \Phi \in \mathcal{H}(h)$, we have $(g \circ \Phi)(\boldsymbol{x}) = h(\boldsymbol{x}) = (h' \circ \psi^{-1})(\boldsymbol{x}), \forall \boldsymbol{x} \in \mathcal{X}$, which amounts to $g(\boldsymbol{z}) = (h' \circ T^{-1})(\boldsymbol{z}), \forall \boldsymbol{z} \in \{-1, 1\}^d$.

For every $h_1, \ldots, h_n, g : \mathcal{Z} \to \{-1, 1\}^n$, and $\Phi : \mathcal{X} \to \mathcal{Z}$ such that $g_i \circ \Phi \in \mathcal{H}(h_i)$ for every $i \in [n]$, we have

$$\lim_{n \to \infty} \frac{1}{n} \widehat{\deg}(g \circ \Phi) = \lim_{n \to \infty} \left( \frac{1}{n} \sum_{j \in [d]} \deg(\Phi_j) + \frac{1}{n} \sum_{i \in [n]} \deg(g_i) \right). \tag{43}$$

Applying Lemma E.1, we have $\deg(\Phi)_j \leq d, \forall j \in [d]$. This gives

$$\lim_{n \to \infty} \frac{1}{n} \sum_{j \in [d]} \deg(\Phi_j) = 0. \tag{44}$$

Meanwhile, since $h_i : \mathcal{X} \to \mathbb{R}, i \in [n]$ are independently and uniformly sampled from $\mathcal{F}^d \circ \psi^{-1}$, we have

$$\lim_{n\to\infty} \frac{1}{n} \sum_{i\in[n]} \deg(g_i) = \mathbb{E}_{h\sim U(\mathcal{F}^d\circ\psi^{-1})}\deg(h \circ f \circ T^{-1}) \tag{45}$$

$$= \mathbb{E}_{h'\sim U(\mathcal{F}^d)}\deg(h' \circ T^{-1}) \tag{46}$$

$$= \frac{1}{|\mathcal{F}^d|} \sum_{h'\in\mathcal{F}^d} \deg(h' \circ T^{-1}). \tag{47}$$

By Lemma E.7, we have $\mathcal{F}^d \circ T^{-1} = \mathcal{F}^d$. This gives

$$\sum_{h'\in\mathcal{F}^d} \deg(h' \circ T^{-1}) = \sum_{h'\in\mathcal{F}^d} \deg(h'). \tag{48}$$

Plugging equations (44), (47), and (48) into equation (43) gives

$$\lim_{n\to\infty} \frac{1}{n} \widehat{\deg}(g \circ \Phi) = \frac{1}{|\mathcal{F}^d|} \sum_{h'\in\mathcal{F}^d} \deg(h'), \tag{49}$$

which is a constant independent of $T$ (and thus independent of $\Phi$). Therefore, for any two viable representations $\Phi, \Phi'$ and $g, g' \in (\mathcal{F}^d)^n$ with $g_i \circ \Phi$ and $g_i' \circ \Phi', \forall i \in [n]$, we must have $\lim_{n\to\infty} \frac{1}{n}\left(\widehat{\deg}(g\circ\Phi) - \widehat{\deg}(g'\circ\Phi')\right) = 0$. This completes the proof. $\qquad\square$

### E.5. Proof of Corollary 4.7

*Proof.* By Definition 3.6, we have

$$\deg(h \mid \psi^{-1}) > 0 \iff \deg(\mathcal{H}_{\min}(h)) > \deg(g) \tag{50}$$

for $g \circ \psi^{-1} \in \mathcal{H}(h)$. By Lemma E.1, we have $\deg(\mathcal{H}_{\min}(h)) \leq d$ for every $h : \mathcal{X} \to \mathbb{R}$. Thus, for $\deg(\mathcal{H}_{\min}(h)) > \deg(g)$ to hold, we must have $\deg(h \circ \psi) = \deg(g) \leq d - 1$. By Definition 4.6, this gives $h \in \mathcal{F}_{d-1}^d \circ \psi^{-1}$, completing the proof. $\qquad\square$

### E.6. Proof of Theorem 4.8

*Proof.* We first prove the following lemma that characterizes the averaged degree change for Boolean function in $\mathcal{F}_k^d$ when composed with invertible transforms.

**Lemma E.10.** *For every integer $1 \leq k \leq d$ and every bijection $T : \{-1,1\}^d \to \{-1,1\}^d$, we have*

$$\sum_{h'\in\mathcal{F}_k^d} \deg(h' \circ T) \geq \sum_{h'\in\mathcal{F}_k^d} \deg(h'). \tag{51}$$

*In particular, when $k = 1$, the equality holds if and only if $\deg(T_i) = 1$ for every $i \in [d]$.*

*Proof of Lemma E.10.* For every $\mathcal{G} \subseteq \mathcal{F}^d$, let $\mathcal{G} \circ T = \{h' \circ T \mid h' \in \mathcal{G}\}$. By Lemma E.7, we know that the mapping $h' \mapsto h' \circ T$ is bijective on $\mathcal{F}^d$. We thus have $|\mathcal{F}_k^d \circ T| = |\mathcal{F}_k^d|$ for every $k$. For $\mathcal{F}_k^d \circ T$, there are two possibilities:

1. $\mathcal{F}_k^d \circ T = \mathcal{F}_k^d$. This immediately gives

$$\sum_{h'\in\mathcal{F}_k^d} \deg(h' \circ T) = \sum_{h'\in\mathcal{F}_k^d\circ T} \deg(h') = \sum_{h'\in\mathcal{F}_k^d} \deg(h'). \tag{52}$$

2. $\mathcal{F}_k^d \circ T \neq \mathcal{F}_k^d$. We can then decompose $\sum_{h'\in\mathcal{F}_k^d} \deg(h' \circ T)$ as follows:

$$\sum_{h'\in\mathcal{F}_k^d} \deg(h' \circ T) = \sum_{h'\in(\mathcal{F}_k^d\circ T)\cap\mathcal{F}_k^d} \deg(h') + \sum_{h'\in(\mathcal{F}_k^d\circ T)\cap(\mathcal{F}_d\backslash\mathcal{F}_k^d)} \deg(h') \tag{53}$$

$$= \sum_{h'\in\mathcal{F}_k^d} \deg(h') + \sum_{h'\in(\mathcal{F}_k^d\circ T)\cap(\mathcal{F}^d\backslash\mathcal{F}_k^d)} \deg(h') - \sum_{h'\in\mathcal{F}_k^d\backslash(\mathcal{F}_k^d\circ T)} \deg(h') \tag{54}$$

Note that $|\mathcal{F}_k^d \circ T| = |\mathcal{F}_k^d|$ gives $|(\mathcal{F}_k^d \circ T) \cap (\mathcal{F}^d \setminus \mathcal{F}_k^d)| = |\mathcal{F}_k^d \setminus (\mathcal{F}_k^d \circ T)|$. Meanwhile, by Definition 4.6, we have $\deg(h') \leq k$ for every $h' \in \mathcal{F}_k^d$ and $\deg(h') > k$ for every $h' \in \mathcal{F}^d \setminus \mathcal{F}_k^d$. Taking these two facts together, we have

$$\sum_{h' \in (\mathcal{F}_k^d \circ T) \cap (\mathcal{F}^d \setminus \mathcal{F}_k^d)} \deg(h') - \sum_{h' \in \mathcal{F}_k^d \setminus (\mathcal{F}_k^d \circ T)} \deg(h') > 0. \tag{55}$$

Plugging equation (55) into equation (54) then gives $\sum_{h' \in \mathcal{F}_k^d} \deg(h' \circ T) > \sum_{h' \in \mathcal{F}_k^d} \deg(h')$.

Combining the above two cases, we conclude that $\sum_{h' \in \mathcal{F}_k^d} \deg(h' \circ T) \geq \sum_{h' \in \mathcal{F}_k^d} \deg(h')$ for every $1 \leq k \leq d$. Note that the above analysis also gives a necessary and sufficient condition for the equality to hold: $\mathcal{F}_k^d \circ T = \mathcal{F}_k^d$.

In particular, when $k = 1$, $\sum_{h' \in \mathcal{F}_k^d} \deg(h' \circ T) = \sum_{h' \in \mathcal{F}_k^d} \deg(h')$ holds only for $T$ satisfying that $\mathcal{F}_1^d \circ T = \mathcal{F}_1^d$. Note that for every non-constant function $h' \in \mathcal{F}_1^d$, there exists $i \in [d]$ such that $(h' \circ T)(\mathbf{z}) \in \{T_i(\mathbf{z}), -T_i(\mathbf{z})\}$ for every $\mathbf{z} \in \{-1,1\}^d$. Due to the arbitrariness of $h'$, we must have $\deg(T_i) = 1$ for every $i \in [d]$. $\qquad\square$

We now move on to prove Theorem 4.8. Our aim is to prove that the minimizer $(\Phi^*, g^*)$ of the optimization problem (10) learns the world model by negations and permutations when the number of tasks $n \to \infty$. Due to equations (43) and (44), the original problem equals to

$$\min_{\Phi: \mathcal{X} \to \mathcal{Z}, g \in \mathcal{F}^d} \lim_{n \to \infty} \frac{1}{n} \sum_{i \in [n]} \deg(g_i) \tag{56}$$

$$\text{s.t.} \quad g_i \circ \Phi \in \mathcal{H}(h_i), \forall i \in [n].$$

Due to the constraint $g_i \circ \Phi \in \mathcal{H}(h_i), \forall i \in [n]$, we know that there must exist a bijection $T: \mathcal{Z} \to \mathcal{Z}$ such that for every $\mathbf{x} \in \mathcal{X}$, $\Phi(\mathbf{x}) = T(\mathbf{z})$, with $\mathbf{z} = \psi^{-1}(\mathbf{x})$ being the true latent variable. Therefore, it remains to prove the existence of a bijection $T: \mathcal{Z} \to \mathcal{Z}$ with $\deg(T_i) = 1, \forall i \in [d]$ such that for every $\mathbf{x} \in \mathcal{X}$, $\Phi^*(\mathbf{x}) = T(\mathbf{z})$.

For every $g_i \circ \Phi \in \mathcal{H}(h_i)$, we have $h_i = g_i \circ \Phi = g_i \circ T \circ \psi^{-1}$. We then have

$$\lim_{n \to \infty} \frac{1}{n} \sum_{i \in [n]} \deg(g_i) = \mathbb{E}_{k \sim \text{Discrete}(p_1, \ldots, p_d)} \mathbb{E}_{h \sim U(\mathcal{F}_k^d \circ \psi^{-1})} \deg(h \circ f \circ T^{-1}) \tag{57}$$

$$= \mathbb{E}_{k \sim \text{Discrete}(p_1, \ldots, p_d)} \mathbb{E}_{h' \sim U(\mathcal{F}_k^d)} \deg(h' \circ T^{-1}) \tag{58}$$

$$= \sum_{k \in [d]} p_k \cdot \frac{1}{|\mathcal{F}_k^d|} \sum_{h' \in \mathcal{F}_k^d} \deg(h' \circ T^{-1}). \tag{59}$$

By Lemma E.10, we have

$$\sum_{h' \in \mathcal{F}_k^d} \deg(h' \circ T^{-1}) \geq \sum_{h' \in \mathcal{F}_k^d} \deg(h') \tag{60}$$

for every $k \in [d]$. Plugging (60) into equation (59) gives

$$\lim_{n \to \infty} \frac{1}{n} \sum_{i \in [n]} \deg(g_i) \geq \sum_{k \in [d]} p_k \cdot \frac{1}{|\mathcal{F}_k^d|} \sum_{h' \in \mathcal{F}_k^d} \deg(h'), \tag{61}$$

where the equality holds only if $\deg(T_i) = 1$ for every $i \in [d]$. This completes the proof. $\qquad\square$

*Remark* E.11. A limitation of Theorem 4.8 is that we requires a non-zero probability of explicitly sampling degree-1 tasks (i.e., $p_1 > 0$), in which latent variables are essentially observed as task outputs. If $p_1 = 0$, we can still prove equation (61) by applying Lemma E.10; in other words, we can still prove that every representation $\Phi^*$ that learns the world model up to negations and permutations is a minimizer of the optimization problem (10). However, such minimizers may not be *unique*, because Lemma E.10 only proves the equivalence between the equality and $\deg(T_i) = 1, \forall i \in [d]$ when $k = 1$ but not $1 < k \leq d - 1$. In fact, we can construct hard examples showing that in some cases, there indeed exists other $\Phi$ that minimizes $\sum_{h' \in \mathcal{F}_k^d} \deg(h' \circ T)$ for every $k \in [d] \setminus \{1\}$.

*Example* E.12. *Let $d = 3$ and let $T : \{-1, 1\}^3 \to \{-1, 1\}^3$ be a bijective transform defined as*

$$T_1(\boldsymbol{z}) = z_1, \quad T_2(\boldsymbol{z}) = z_1 z_2, \quad T_3(\boldsymbol{z}) = z_1 z_3. \tag{62}$$

*One can easily verify that for every $k \in \{2, 3\}$, every parity function $\chi_S(T(\boldsymbol{z})) = \prod_{i \in S} T_i(\boldsymbol{z})$ with $|S| \le k$ satisfy $\deg(\chi_S(T(\boldsymbol{z}))) \le k$. By the Fourier-Walsh transform, this amounts to $\mathcal{F}_k^3 \circ T = \mathcal{F}_k^3$ for $k = \{2, 3\}$, which gives $\sum_{h' \in \mathcal{F}_k^3} \deg(h' \circ T) = \sum_{h' \in \mathcal{F}_k^3} \deg(h')$ by the proof of Lemma E.10. Thus, in this case we require $p_1 > 0$ to ensure that the representation $\Phi$ satisfying $\Phi(\boldsymbol{x}) = T^{-1}(\boldsymbol{z})$ for every $\boldsymbol{x} \in \mathcal{X}$ and $\boldsymbol{z} = \psi^{-1}(\boldsymbol{x})$ is not a minimizer of (10).*

Nevertheless, we do believe that cases like Example E.12 are rare. This is because by the proof of Lemma E.10, such examples must construct a bijection $T$ such that $\mathcal{F}_k^d \circ T = \mathcal{F}_k^d$ for *every* $k \in [d] \setminus \{1\}$, which is increasingly difficult when $d$ becomes large. For example, if we increase the dimension of $\mathcal{Z}$ from 3 to 4 in Example E.12 and keep $T_1$, $T_2$ and $T_3$ as is, it could be verified that there does not exist a $T_4 : \{1, 1\}^3 \to \{-1, 1\}$ satisfying that $\mathcal{F}_k^4 \circ T = \mathcal{F}_k^4$ for every $k \in \{2, 3\}$. We believe that this intuition could be rigorously proved using e.g., Lemma E.6 or other techniques and leave it as future work.

### E.7. Proof of Theorem 4.9

*Proof.* We first prove the following lemma:

**Lemma E.13.** *Assume that the latent variables are uniformly sampled from the Hamming ball $B_r = \{\boldsymbol{z} \in \{\pm 1\}^d \mid \#_{-1}(\boldsymbol{z}) \le r\}$ with $r < d$. Then, for every task $h$, we have*

$$\deg(\mathcal{H}_{\min}(h)) \le \left\lceil \log_2 \sum_{i=0}^{r} \binom{d}{r} \right\rceil. \tag{63}$$

*Proof of Lemma E.13.* The main idea of the proof is similar to that of Lemma E.1 and Lemma E.2.

Let $k = \lceil \log_2 \sum_{i=0}^{r} \binom{d}{r} \rceil$. Due to the invertibility of $\psi$, we know that $|\{\boldsymbol{x} \mid p(\psi^{-1}(\boldsymbol{x})) > 0\}| = |B_r| = \sum_{i=0}^{r} \binom{d}{r}$. Therefore, for any $S = \{i_1, \ldots, i_{k'}\} \subseteq [d]$ such that $k' > k$, there must exist $b_1, \ldots, b_{k'} \in \{-1, 1\}^{k'}$ such that

$$\prod_{j \in [k']} (x_{i_j} + b_j) = \prod_{j \in S} x_j + \sum_{S' \subset S, |S'| \le k'-1} b_{S'} \prod_{k \in S'} x_k = 0 \tag{64}$$

for every $\boldsymbol{x} \in \mathcal{X}' := \{\boldsymbol{x} \mid p(\psi^{-1}(\boldsymbol{x})) > 0\}$, where $b_{S'} \in \{-1, 1\}$ for every $S' \subset S$–if this does not hold, then we have $|\mathcal{X}'| \ge 2^{k'} > 2^k \ge \sum_{i=0}^{r} \binom{d}{r}$, which contradicts $|\mathcal{X}'| = \sum_{i=0}^{r} \binom{d}{r}$. By equation (64), we can replace every degree-$k'$ monomial $\chi_S(\boldsymbol{x}) = \prod_{j \in S} x_j$ by a degree-$k' - 1$ polynomial $-\sum_{S' \subset S, |S'| \le k'-1} b_{S'} \prod_{k \in S'} x_k$. Iteratively using this replacement in the Fourier-Walsh transform of $h$ gives the desired result. $\square$

We can now prove Theorem 4.9.

*Proof of (i).* By Lemma E.13, for every $h^* \in \mathcal{H}_{\min}(h)$, we have $\deg(h^*) \le k = \lceil \log_2 \sum_{i=0}^{r} \binom{d}{r} \rceil$. The test MSE of any $h^*$ thus satisfies

$$\mathrm{err}(h^*) = \mathbb{E}_{\boldsymbol{z} \sim U(\{-1,1\}^d)}[(h^*(\mathbf{x}) - h(\mathbf{x}))^2] = \mathbb{E}_{\boldsymbol{z} \sim U(\{-1,1\}^d)}[h^*(\mathbf{x})^2 + h(\mathbf{x})^2 - 2h^*(\mathbf{x})h(\mathbf{x})] \tag{65}$$

$$= 1 + \mathbb{E}_{\boldsymbol{z} \sim U(\{-1,1\}^d)} h^*(\mathbf{x})^2 - 2 \mathbb{E}_{\boldsymbol{z} \sim U(\{-1,1\}^d)} h^*(\psi(\boldsymbol{z})) h(\psi(\boldsymbol{z})) \tag{66}$$

$$> 1 - 2 \langle h^* \circ \psi, h \circ \psi \rangle \tag{67}$$

We then prove that $k \ge r + 1$. To see this, recall that $d > r$ and one can verify

$$k = \left\lceil \log_2 \sum_{i=0}^{r} \binom{d}{r} \right\rceil \ge \left\lceil \log_2 \sum_{i=0}^{r} \binom{r+1}{r} \right\rceil = \lceil \log_2 (2^{r+1} - 1) \rceil = r + 1. \tag{68}$$

Since $\deg(h \mid \psi^{-1}) \ge q - r$, by Definition 3.6 we have $\deg(h^*) - \deg(g) \ge q - r$ for every $g$ satisfying $g \circ \psi^{-1} = h^*$. This gives $\deg(g) \le \deg(h^*) - q + r \le k - q + r$, i.e., $\deg(h^* \circ \psi) \le k - q + r$. Recalling that $h \circ \psi$ is a parity function $\chi_S$ with $\deg(h \circ \psi) = |S| > k - q + r$ and applying equation (19), we have

$$\langle h^* \circ \psi, h \circ \psi \rangle = \widehat{h^* \circ \psi}(S) = 0. \tag{69}$$

Plugging equation (69) into (67) completes the proof.

*Proof of (ii).* Since $\deg(h \mid \psi^{-1}) \geq q - r$ and $\Phi^*$ learns the world model up to negations and permutations, we have $\deg(h \mid \Phi^*) = \deg(h \mid \psi^{-1}) \geq q - r$ and hence $\deg(g^*) \leq r$. By Lemma E.8, $g^*$ is unique. We thus have $h = g^* \circ \Phi^*$, which gives the desired result. □

### E.8. Proof of Theorem 4.12

*Proof.* We first prove the following lemma:

**Lemma E.14.** *If $U$ is compatible, then $\deg_U(f) = \deg(f)$ holds for every $f \in \mathcal{F}^n$.*

*Proof of Lemma E.14.* By Definition 4.11, we have $\deg(U(\chi_S)) = \deg(\chi_S)$ for every compatible $U$. Applying Lemma E.9, we further have $\deg(U(\chi_S)) = \deg(\chi_S) = \deg(U^{-1}(\chi_S))$. This gives

$$\deg_U(f) = \max\left\{\deg(U^{-1}(\chi_S)) : \hat{f}(S) \neq 0\right\} = \max\left\{\deg(\chi_S) : \hat{f}(S) \neq 0\right\} = \deg(f), \tag{70}$$

which completes the proof. □

We are now ready to prove Theorem 4.12. Note that under the new basis $\{U(\chi_S)\}$, the original optimization problem (10) becomes (also applying the equivalence between (10) and (56)):

$$\min_{\Phi:\mathcal{X}\to\mathcal{Z}, g\in\mathcal{F}^d} \lim_{n\to\infty} \frac{1}{n} \sum_{i\in[n]} \deg_U(g_i) \tag{71}$$
$$\text{s.t.} \quad g_i \circ \Phi \in \mathcal{H}(h_i), \forall i \in [n].$$

*Proof of (i).* If $U$ is compatible, then by Lemma E.14 we have $\deg_U(f) = \deg(f)$ for any Boolean function $f$. This immediately gives the equivalence between (71) and (56) and hence their minimizers. We thus have that $\Phi^*$ learns the world model up to negations and permutations as in Theorem 4.8.

*Proof of (ii).* If $U$ is not compatible, then due to the invertibility of $U$, there exists at least one $h'' \in \mathcal{F}_k^d$ such that $\deg_U(h'' \circ T^{-1}) > \deg(h'')$ for some $k < d$. Since composing Boolean functions with degree-1 transforms does not change the degree of functions, we have $\deg_U(h'' \circ T^{-1}) > \deg(h'')$ and hence $\sum_{h'\in\mathcal{F}_k^d} \deg_U(h' \circ T^{-1}) > \sum_{h'\in\mathcal{F}_k^d} \deg(h')$.

In particular, let $\{\chi_1, \ldots, \chi_{2^d}\}$ be the set of all parity functions with $d$-dimensional inputs. For every $k \in [d]$, we can construct $U$ such that:

1. $U(\chi_1), \ldots, U(\chi_{2^d})$ is a permutation of $\chi_1, \ldots, \chi_{2^d}$;

2. For every $i \in [d]$, we have $U(\chi_{\{i\}}) = \chi_S$ for some $S \subseteq [d]$ with $|S| = k$ and $U(\chi_S) = \chi_{\{i\}}$.

Recall that $\deg_U(h' \circ T^{-1}) = \deg(U^{-1}(h' \circ T^{-1}))$. Therefore, to ensure that $\sum_{h'\in\mathcal{F}_k^d} \deg_U(h' \circ T^{-1}) = \sum_{h'\in\mathcal{F}_k^d} \deg(h')$ for $k = 1$, we thus must have $T_j^{-1} = \chi_S$ for some $S \subseteq [d]$ and $j \in [d]$, with $|S| = k$. This means

$$\max_{i\in[d]} \deg\left(T_i^{-1}\right) \geq \deg\left(T_j^{-1}\right) = k, \tag{72}$$

which gives the desired result. □

## F. Experimental Details

This section presents additional experimental details. All of our experiments were conducted using PyTorch (Paszke et al., 2019) on NVIDIA V100/A100 GPUs.

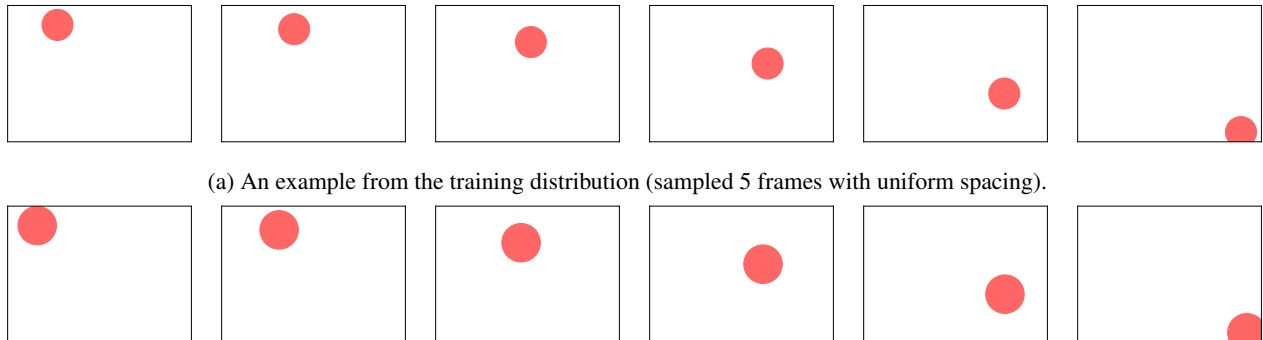

(a) An example from the training distribution (sampled 5 frames with uniform spacing).

(b) An example from the test distribution with a larger radius and a larger initial velocity (sampled 5 frames with uniform spacing).

*Figure 6.* Two visualized examples of the parabolic motion.

### F.1. Polynomial Extrapolation

**Dataset.** We consider fitting and extrapolating degree-$n$ polynomials with the form $P_n(x) = \sum_{i=0}^{n} a_i x^n$. Given an input $x \in \mathbb{R}$, the label is given by $y = P_n(x)$. In our experiments, we consider three families of polynomials with degree 1, 2, and 3. In each family, every coefficient $a_i, i \in \{0, 1, \ldots, n\}$ is uniformly sampled from $[0, 1)$. In our experiments, we sample 50 polynomials in each family for the violin plots. Other data parameters are as follows:

- Training, validation, and test data are uniformly sampled from $[-1, 1)$, $[-1, 1)$, and $[-2, 2)$, respectively.

- For each polynomial instance, we sample $50,000$ training data, $1,000$ validation data, and $10,000$ test data.

**Model and hyperparameters.** We consider MLPs with the following architecture:

$$\text{MLP}(\boldsymbol{x}) = \boldsymbol{W}^{(d)} \sigma \left( \boldsymbol{W}^{(d-1)} \sigma \left( \ldots \sigma \left( \boldsymbol{W}^{(1)} \boldsymbol{x} + \boldsymbol{b}^{(1)} \right) \right) + \boldsymbol{b}^{(d-1)} \right) + \boldsymbol{b}^{(d)}, \tag{73}$$

where $\sigma$ is the activation function and for every $i \in [d]$, $\boldsymbol{W}^{(i)}$ and $\boldsymbol{b}^{(i)}$ are weights and bias of the $i$-th layer, respectively. For ReLU MLPs, all activation functions are set to ReLU; for our architecture, we replace half of ReLUs in every layer by the identity function $\sigma(x) = x$ and the quadratic function $\sigma(x) = x^2$, with the number of identity functions and quadratic functions being the same (i.e., both functions constitute $25\%$ activation functions, while the remaining $50\%$ are still ReLUs). We search the following hyperparameters for MLPs:

- Number of layers $d$ is set to 4.

- Width of each $\boldsymbol{W}^{(i)}$ from $\{128, 256, 512\}$.

We train all MLPs with the mean square error (MSE) loss with the AdamW optimizer (Loshchilov & Hutter, 2019). Training hyperparameters are as follows:

- Initial learning rate from $\{1e-3, 1e-4, 1e-5\}$. We use a cosine learning rate scheduler.

- Weight decay is set to $0.1$.

- Batch size is set to $512$.

- Number of epochs is set to $400$.

**Evaluation metric.** We evaluate all models using MSE on test data.

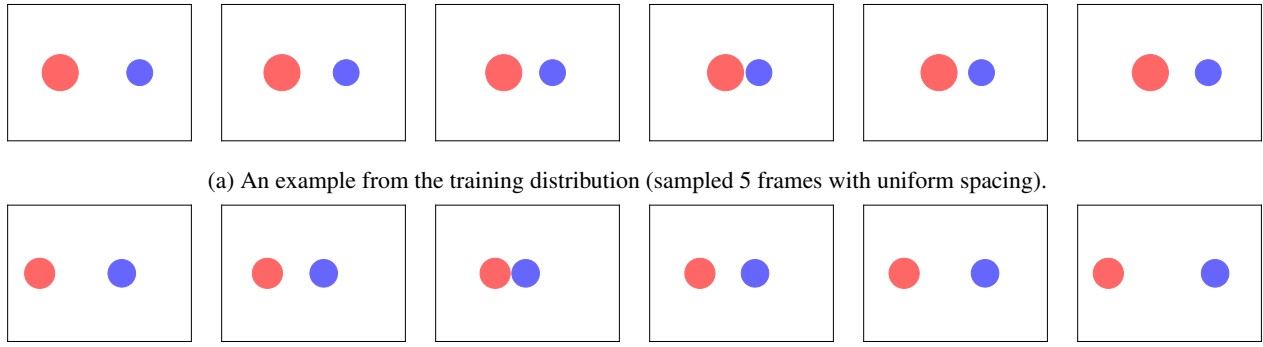

(a) An example from the training distribution (sampled 5 frames with uniform spacing).

(b) An example from the test distribution with larger initial velocities (sampled 5 frames with uniform spacing).

*Figure 7.* Two visualized examples of the collision motion.

### F.2. Learning Physical Laws

**Dataset.**  Inspired by Kang et al. (2025), we create training and test sequences representing ball-shaped object movements that adhere to two basic physical laws: (i) single-object parabolic motion (reflecting Newton's second law of motion), and (ii) two-object elastic collision (reflecting the convervation of energy and momentum). In both settings, we consider a 2-dimensional environment in which each object is encoded by a three-dimensional tuple $(r_t, x_t, y_t)$ at every time step $t$, where $r_t$ represents the radius of the ball and $(x_t, y_t)$ stands for the 2-dimensional coordinates of the ball. To encode the timestamp information, we also include a dimension for the current timestamp $t$. This results in 4-dimensional inputs (i.e., $(r_t, x_t, y_t, t)$) in the parabolic motion setting and 7-dimensional inputs (i.e., $(r_t^1, x_t^1, y_t^1, r_t^2, x_t^2, y_t^2, t)$ in the collision motion setting. Each motion sequence consists of 32 frames with a timestep of 0.1. Details of both settings are as follows.

1. **Parabolic motion.** This motion describes the process where a ball with an initial horizontal (i.e., along the $x$ axis) velocity falls due to a fixed gravity $g = 9.8$ (along the $y$ axis). We use the following training and test parameters:

   - Radius is uniformly sampled from $[0.7, 1.5]$ in training and from $[1.5, 2.0]$ in test.
   - Initial velocity is uniformly sampled from $[1, 4]$ in training and from $[4.5, 6.0]$ in test.

2. **Collision motion.** This motion describes the process where two balls with different sizes and different initial velocities move horizontally towards each other and collide. We assume that the collision is perfectly elastic and all balls are with the same density, so the velocities of both balls can be inferred from their radii and initial velocities. We use the following training and test parameters:

   - Radius of each object is uniformly sampled from $[0.7, 1.5]$ in training and from $[1.5, 2.0]$ in test.
   - Initial velocity is uniformly sampled from $[2, 4]$ in training and from $[4.5, 6.0]$ in test.
   - The horizontal distance between two objects is uniformly sampled from $[5, 15]$ in both training and test.

See Figure 6 and Figure 7 for visualized examples of the parabolic motion and examples of the collision motion.

For both settings, we sample 1M training sequence and $50,000$ test sequence.

**Model and hyperparameters.**  We train a decoder-only transformer (Vaswani et al., 2017) conditioned on the first 3 frames to predict the remaining frames (expect for the timestamp dimension). For our method, we simply replace every MLP in the original transformer with our modified MLP as in Section F.1 (replacing ReLU by GELU). We use teacher-forcing in training that is similar to next-token prediction, i.e., the model only needs to predict the next frame (starting from the 4-th frame to the last 32-th frame) given all ground-truth frames prior to it. We use the following model hyperparameters:

- Number of layers of transformer is set to $4$.

- Number of heads of transformer is set to $4$.

- Width of transformer is set to $512$.

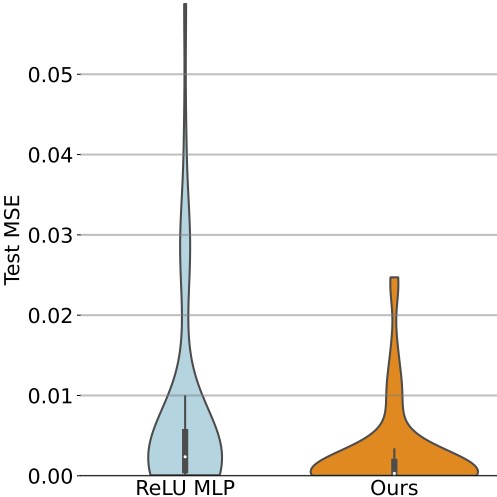

*Figure 8.* Violin plots of the test MSE of the ReLU MLP and our model in extrapolating degree-1 polynomials.

We train all models using the MSE loss with the AdamW optimizer. Training hyperparameters are as follows:

- Initial learning rate is randomly sampled from $[1e - 6, 1e - 3]$. We use a cosine learning rate scheduler.

- Weight decay is set to $1e - 4$.

- Batch size is set to $1024$.

- Number of epochs is set to $300$.

**Evaluation metric.** For test, we iteratively use the trained model to predict all missing frames given the first 3 frames. The predicted frames will be used together with the given frames for the model to predict the next frame. We evaluate all models using MSE on test data, averaged over all predicted 29 frames for each sequence.

## G. Additional Results and Discussion

This section presents additional empirical results and discussion.

### G.1. Polynomial Extrapolation

In the main text, we report extrapolation results on degree-2 and degree-3 polynomials in Figure 2b; for completeness, here we also report extrapolation results on degree-1 polynomials, i.e., linear functions. As shown by (Xu et al., 2021), ReLU MLPs can also extrapolate well in this setting. The violin plots of the test MSE of both the ReLU MLP and our model are in Figure 8. We can see that while both models achieve a much smaller extrapolation error compared to those in extrapolating higher degree polynomials, our model still outperforms the ReLU MLP. We also provide more examples for extrapolating degree-1, degree-2, and degree-3 polynomials in Figure 9, Figure 10, and Figure 11, respectively.

**Comparison with Xu et al. (2021).** While Xu et al. (2021) also show that 2-layer MLPs with quadratic activation functions can extrapolate quadratic functions better than ReLU MLPs, we emphasize that there are two key differences between our results and theirs:

1. Xu et al. (2021) replaces *all* ReLU functions with quadratic activation functions, while we only replace half of ReLU functions with quadratic activation functions and identity activation functions. This difference is important in scenarios where we do not know the exact structural form of target functions–note that our method can be viewed as an "ensemble" of different activation functions, which enables the neural network to adaptively select activation functions that are the most compatible with the task as shown by our empirical results.

2. Xu et al. (2021) only considers 2-layer MLPs for learning quadratic functions, while we consider using 4-layer MLPs for learning degree-2 and degree-3 polynomials. This difference enables us to verify that neural networks can learn bases that require *function compositions*. For example, degree-3 polynomials need a basis function $y = x^3$, which cannot be composed using a 2-layer MLP but is composable using MLPs with more than two layers and with quadratic and identity activation functions. We also note that the inclusion of identity functions is important since MLPs with only quadratic activations can only represent basis functions $y = x^k$ with even degrees $k$.

## G.2. Learning Physical Laws

We present visualization results of the transformer baseline and our model in Figure 12 in a test collision motion example. We can see that while both models yield accurate predictions before the collision, our model outperforms the baseline on the predictions of the object velocities after the collision.

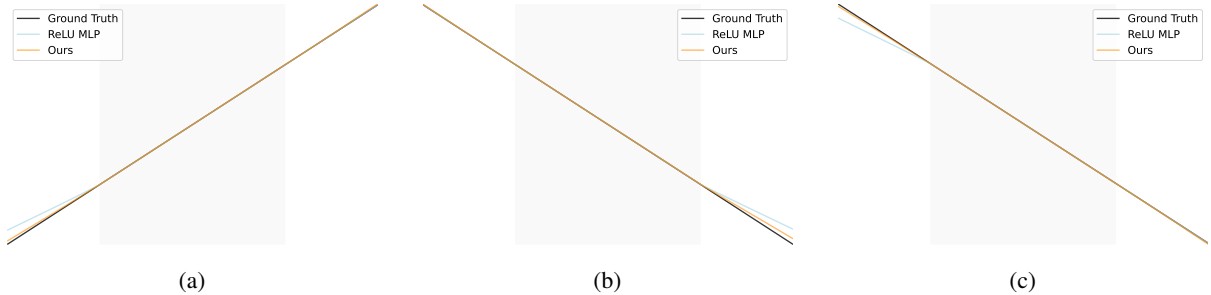

*Figure 9.* Selected examples for degree-1 polynomial extrapolation. Shaded regions indicate training regions.

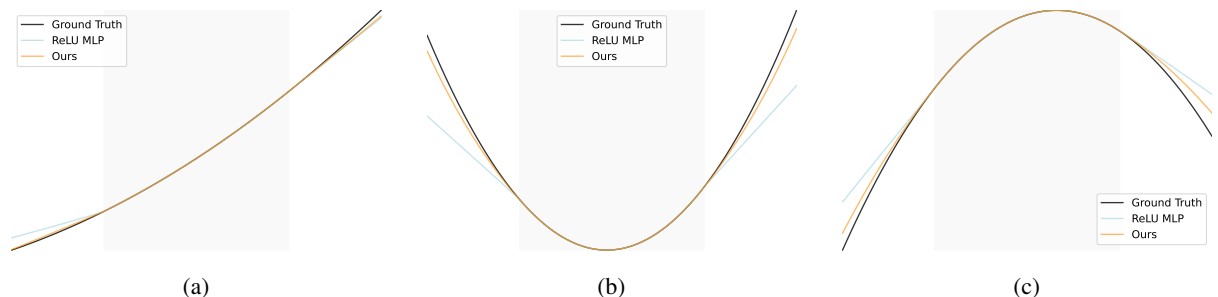

*Figure 10.* Selected examples for degree-2 polynomial extrapolation. Shaded regions indicate training regions.

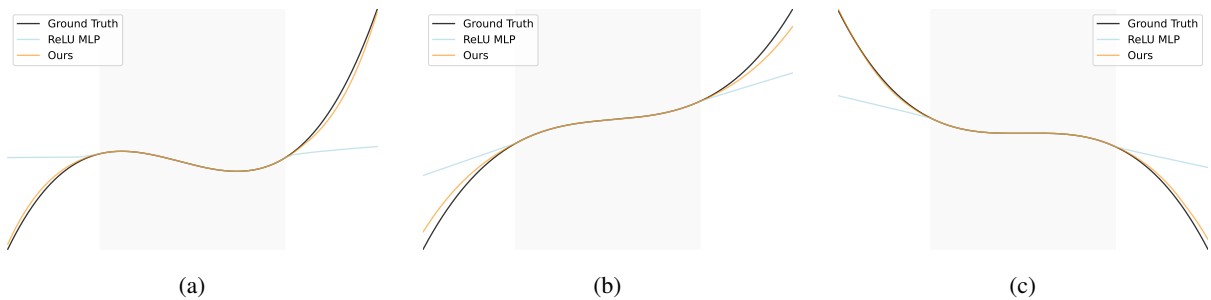

*Figure 11.* Selected examples for degree-3 polynomial extrapolation. Shaded regions indicate training regions.

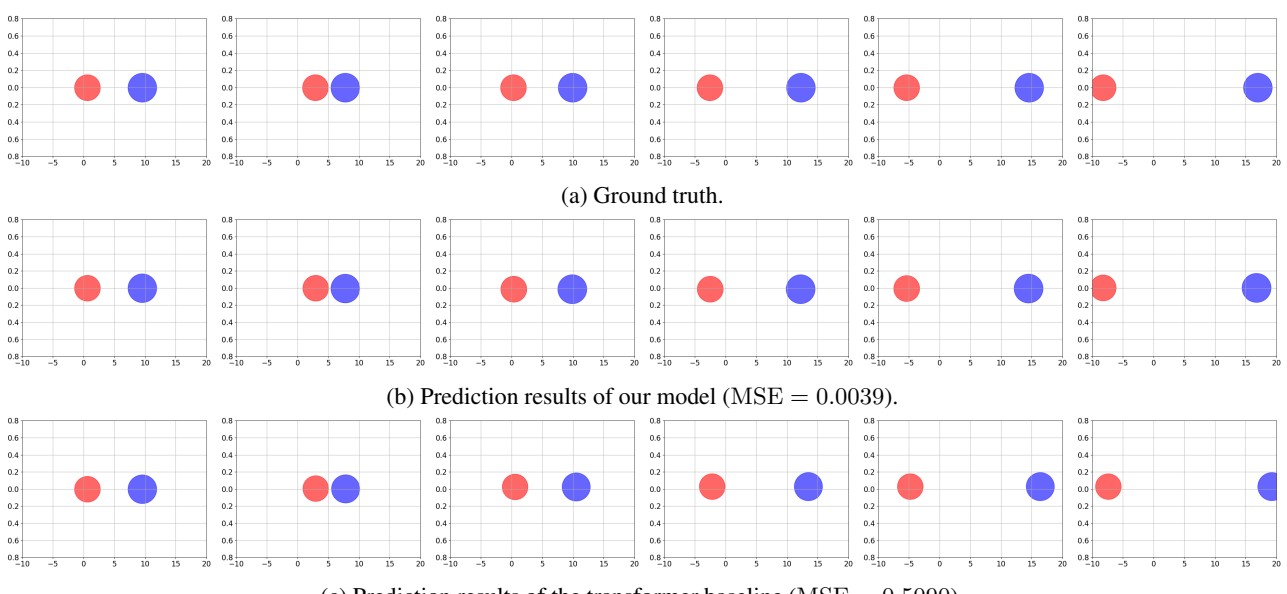

(a) Ground truth.

(b) Prediction results of our model (MSE = 0.0039).

(c) Prediction results of the transformer baseline (MSE = 0.5090).

*Figure 12.* Visualization results in a test collision motion example. All three rows select the same frames with uniform spacing.

