# OpenReview forum: "When do neural networks learn world models?"
_ICML.cc/2025/Conference — ICML 2025 poster_

### Official Review · Reviewer_L9Q8 · 2025-02-24

**Overall Recommendation:** 5

**Summary:**

The paper explores whether neural networks can develop internal world models akin to those humans create to understand the underlying data generation processes. The research presents the first theoretical results in this area, demonstrating that in a multi-task setting, models with low-degree bias can recover latent data-generating variables under mild assumptions. This recovery, however, is sensitive to model architecture. Using Boolean models of task solutions via the Fourier-Walsh transform, the researchers introduce new techniques to analyze invertible Boolean transforms. They show that the implicit bias in deep learning towards low-complexity functions can favor certain solutions over others. Their findings have implications for self-supervised learning, out-of-distribution generalization, and the linear representation hypothesis in large language models. The paper provides a formal framework for world model learning and presents theoretical results, identifying key factors such as multi-task settings and low-complexity bias necessary for the successful learning of world models. The research also illustrates algorithmic implications through tasks like polynomial extrapolation and learning physical laws, showing that architectures informed by their analysis can outperform conventional ones.

**Claims And Evidence:**

The claims are well supported by the theorems. They could be better supported if the authors added some experiments.

**Essential References Not Discussed:**

No

**Experimental Designs Or Analyses:**

I think additional experiments should be added to better support the main claim.

**Methods And Evaluation Criteria:**

No proposed method

**Other Comments Or Suggestions:**

No

**Other Strengths And Weaknesses:**

**Strengths:**

1. This is pioneering work in studying the data generation model when learned by neural networks.

2. Most assumptions are realistic and facilitate the proof of the results.

3. Theorems 4.3 and 4.8 are novel to me and are of practical importance.

**Weaknesses:**

1. The major weakness is that the experiments are not strongly related to the theorems. I would improve the score to "strong accept" if this point is addressed. (Please refer to Questions 1, 2, 3.)

2. "World model" in the title is defined as the data generation model. This does not seem to be a widely accepted term. The authors should support this claim with more existing works or consider changing the title.

**Questions For Authors:**

1. A major assumption is that the neural network will learn the function with minimal (realization) degree (e.g., (ii) in Theorem 4.8). The authors support this assumption with literature on "simplicity bias," but I believe that additional support from synthetic experiments is important.

2. Theorems 4.1 and 4.3 state that the data generation model $\Phi$ emerges if we adopt a multi-task learning scheme. Could the authors support this claim with synthetic experiments?

3. Could Theorems 4.8 and 4.9 be supported with synthetic experiments?

**Relation To Broader Scientific Literature:**

This is pioneering work in studying the data generation model when learned by neural networks. Most assumptions are realistic and facilitate the proof of the results. Theorems 4.3 and 4.8 are novel to me and are practically important. Therefore, I believe this paper will make a significant impact in this field in the following ways: the assumptions can be reused for other theoretical studies, and the conclusions can inspire future empirical work.

**Theoretical Claims:**

I haven't checked every detail of the proof due to time constraints, but I believe the logic of the proof is correct.

---

> ### Author Rebuttal · Authors · 2025-04-01
>
> Thank you for your detailed review and your approval of our work. We provide point-by-point responses to your questions in the following:
>
> **Synthetic experiments:** Thank you for these questions; we believe that they greatly help to improve our work. Inspired by them, we conducted additional synthetic experiments using Boolean inputs and MLPs to further support our theoretical claims. _Please find our experimental results in this anonymous link: https://anonymous.4open.science/r/rebuttal_figures-8C6A/rebuttal_figures.pdf._ (Backup links: https://ibb.co/mrdwJt9T and https://ibb.co/gNX4FYT if the former link is not available). A brief introduction on how these results answer your questions is as follows:
>
>   - **Low-degree bias of neural networks (Q1):** _Figure 1_ shows that MLPs indeed have a low-degree bias when solutions with different degrees exist for the same training distribution.
>
>   - **Multi-task learning and the identification of data-generating variables (Q2 & Q3, Theorem 4.3 & Theorem 4.8):** _Figure 2_ shows that the identification error of data-generating variables significantly decreases as more training tasks are drawn, while single-task learning does not lead to similar identification.
>
>   - **Identifying data-generating variables benefits OOD generalization (Q3, Theorem 4.9):** _Figure 3_ shows that in a Hamming sampling setting as in Theorem 4.9, MLPs with a representation $\Phi$ that identifies true data-generating variables achieve superior OOD generalization compared to MLPs without such a representation.
>
> We will add these experiments to the next version of our paper.
>
> **Definition of the world model:** We believe that defining the world model as a model that learns data-generating variables, although not formally established as a standard term, is well-motivated by the existing literature:
>
>   - In the first work that introduces the term "world model" in the machine learning context [1], the authors describe the world model as learning "an abstract representation of both spatial and temporal aspects" of the information flow, which naturally motivates a representation learning formulation. The authors of [1] also experimentally demonstrate this concept by training an RNN-based _generation_ model on top of a learned _latent vector_, which involves latent data generation models. Thus, our definition can be viewed as a natural abstraction of their model.
>
>   - Many existing works explicitly use very similar definitions as ours: For example, the authors of [2] wrote "An alternative hypothesis is that LLMs, in the course of compressing the data, learn more compact, coherent, and interpretable models of the _generative process_ underlying the training data, i.e., a world model", while [3] defines a "causal world model" as "a causal model of the _data generating process_".
>
>   - A series of empirical works (e.g., [4-8]) also share a similar spirit: [4-6] treat world models as "understandable models of the process producing the sequences they (language models) are trained on" [4]; empirically, they test if transformers learn world models/world representations by connecting the models' internal representations to environment states that generate the training sequence. [7] adopts a similar approach to the task of generating program traces. [8] describes "linear world models" as models with high-level features linearly decodable from their latent spaces (see also our discussion "Connection to the linear representation hypothesis" in Section 4.2).
>
> Given the above discussion, we feel that our current title is consistent with the related literature. We will also include more discussion in the next version of our paper.
>
> **Please let us know if you still have any questions or concerns. We are happy to engage in further discussion.**
>
> ---
>
> [1] Ha and Schmidhuber. World models. NeurIPS, 2018.
>
> [2] Gurnee and Tegmark. Language models represent space and time. ICLR, 2024.
>
> [3] Richens and Everitt. Robust agents learn causal world models. ICLR, 2024.
>
> [4] Li et al. Emergent world representations: Exploring a sequence model trained on a synthetic task. ICLR, 2023.
>
> [5] Nanda et al. Emergent linear representations in world models of self-supervised sequence models. EMNLP, 2023.
>
> [6] Toshniwal et al. Learning chess blindfolded: Evaluating language models on state tracking. AAAI, 2022.
>
> [7] Jin and Rinard. Emergent representations of program semantics in language models trained on programs. ICML, 2024.
>
> [8] Marks and Tegmark. The geometry of truth: Emergent linear structure in large language model representations of true/false datasets. COLM, 2024.

---

> > ### Comment · Reviewer_L9Q8 · 2025-04-02
> >
> > Most of my concerns have been addressed and I have raised the score. I enjoy reading the paper. Could you further explain in detail how you identify the data generation variable? For example, (1) why use linear probing instead of a direct feature selection? (2) which layer is suitable for linear probing?

---

> > > ### Author Response · Authors · 2025-04-08
> > >
> > > Thank you for your engagement in the discussion! We are glad that our responses have addressed most of your concerns. In the following, we provide more details on our data-generating variable identification experiment (Figure 2 in the link) to hopefully answer your follow-up questions. We will also add more experimental details and discussion to the next version of our paper.
> > >
> > > In the experiment, we trained a shared three-layer MLP as the representation $\Phi$ and $n$ separate three-layer MLPs as task-specific decoders $g_i,i\in[n]$ given $n$ tasks, following our theoretical setup. For data-generating variable identification, we used the learned representation $\Phi(x)$, i.e., the final-layer output of the representation MLP. For practical neural nets, $\Phi(x)$ is a float vector rather than Boolean, so we did not perform direct feature selection and instead used linear probing, which amounts to examining whether the true latent variables $z$ can be identified up to a linear transform. This is consistent with Theorem 4.8, which shows that $z$ can be identified up to degree-1 transforms in the Boolean domain (corresponding to linear transforms in the real domain). Note that similar linear probing techniques have also been used in prior empirical works on world models (e.g., [1-3]).
> > >
> > > ---
> > >
> > > [1] Nanda et al. Emergent linear representations in world models of self-supervised sequence models. EMNLP, 2023.
> > >
> > > [2] Gurnee and Tegmark. Language models represent space and time. ICLR, 2024.
> > >
> > > [3] Marks and Tegmark. The geometry of truth: Emergent linear structure in large language model representations of true/false datasets. COLM, 2024.

---

### Official Review · Reviewer_GtGz · 2025-03-06

**Overall Recommendation:** 3

**Summary:**

this paper aims to answer the question "when do neural nets learn world models". This is an ambitious goal that has been studied in many papers. the paper proposes a theoretical results for this problem. then the paper also shows the algorithmic implications of the results.

**Claims And Evidence:**

yes

**Essential References Not Discussed:**

references are adequate

**Experimental Designs Or Analyses:**

yes

**Methods And Evaluation Criteria:**

yes

**Other Comments Or Suggestions:**

n/a

**Other Strengths And Weaknesses:**

i believe causality is important in world models, but is not explored in this paper.

I'm not very familiar with the content in this paper so I'll reevaluate the paper after reading the comments from the other reviewers.

**Questions For Authors:**

please see the comments above

**Relation To Broader Scientific Literature:**

i believe this paper will have some impact to a sizable audience in the ML community

**Theoretical Claims:**

yes

---

> ### Author Rebuttal · Authors · 2025-04-01
>
> Thank you for your review. We are happy to engage in further discussion if you have additional questions or concerns in the author-reviewer discussion stage.
>
> On the point of causality, we agree that causality is also an important aspect of learning world models. Moreover, we would like to emphasize that a well-known difficulty in causal discovery is the identification of causal latent variables when the observed data is generated from them in a _non-linear_ fashion (see e.g., [1]), for which we show that such identification could be hoped if the model has a low-complexity bias. Therefore, combining our results and the results in causal discovery is interesting future work.
>
> ---
>
> [1] Schölkopf et al. Toward causal representation learning. Proceedings of the IEEE, 2021.

---

### Official Review · Reviewer_i1Ec · 2025-03-10

**Overall Recommendation:** 3

**Summary:**

This paper provides a theoretical framework to reveal whether neural networks are capable of learning world models that capture the underlying data generation process. The framework connects the latent variable models with the world model learning. The theoretical results provide insights for future research.

## update after rebuttal
Thanks to the authors for providing the rebuttal. I've read the author's response and comments from other reviewers. I have no further questions at this time. I will keep my original positive rating.

**Claims And Evidence:**

The theoretical framework and results support the paper's claims.

**Essential References Not Discussed:**

To the best of my knowledge, the references are sufficiently covered.

**Experimental Designs Or Analyses:**

The experimental results support the claims.

**Methods And Evaluation Criteria:**

This paper mainly proposes a theoretical framework and experiments on polynomial extrapolation and  learning physical
laws.

**Other Comments Or Suggestions:**

N/A

**Other Strengths And Weaknesses:**

[Strengths]
1. The overall paper is solid and well-structured.
2. The theoretical results are comprehensive and insightful.
3. In addition, the experimental results are provided in this paper to further verify the claims of this paper and help readers understand the proposed concepts more easily.

Overall, this paper is solid and provides insightful observations/proof for future research in the areas of physical AI, robotics, autonomous driving, etc. Despite some questions listed in the following Questions For Authors section, no major technical concerns are found at this time.

**Questions For Authors:**

1. Whether the derived insights could be applied to model-based RL tasks?
2. Considering that video generation-based world models play crucial roles in robotics and autonomous driving, I'm wondering whether the derived insights could benefit the design of large video world models by allowing them to capture real-world physics from data.

**Relation To Broader Scientific Literature:**

This paper is among the first to propose a theoretical framework to study the problem of when neural networks learn world models.

**Theoretical Claims:**

I have checked the main theoretical claims, but not in every detail.

---

> ### Author Rebuttal · Authors · 2025-04-01
>
> Thank you for your review and your approval of our work. The following are our responses to your questions.
>
> **Applications:** Thank you for your questions on potential algorithmic applications of our results. Yes, we believe that our results may inspire future algorithmic research in the following ways.
>  - The main message of Theorem 4.3 is that the conditional degree of proxy tasks is important for representation learning. This may inspire the development of self-supervised representation learning objectives by maximizing their conditional degree (e.g., pixel-level prediction may not be a good objective since it potentially has a small conditional degree), which could be applied to designing auxiliary/proxy tasks in model-based RL, training video generation models, and language modeling. We are now exploring this idea further in the context of training language models by augmenting next-token prediction with auxiliary training objectives.
>  - The notion of basis compatibility shows that accommodating the inductive biases of the proxy tasks in the network architecture can benefit world model learning, as demonstrated in our proof-of-concept experiments. This may inspire future model design for learning and extrapolating physical laws in a larger context. For example, for video generation models to robustly capture periodicity in real-world physics, we may need to explicitly embed modules with a natural periodicity behavior, rather than merely hoping that models with standard architectures could learn such behaviors from data.
> - More broadly, an important takeaway of our work is that "while neural networks typically exhibit 'simplicity biases', the underlying meaning of 'simplicity' is not fixed and is _dependent_ on the network architecture". To our knowledge, this is an underexplored aspect in the existing literature on the simplicity bias of neural networks. This could also explain why the simplicity bias tends to be sometimes beneficial and sometimes harmful in different tasks. Moving forward, we believe that designing network architectures that could _adaptively_ fit such simplicity biases to the "right" inductive biases of training tasks is a promising direction. We are currently working on a project on this topic.
>
> We will also include more discussion in the next version of our paper.
>
> **Please let us know if you still have any questions or concerns. We are happy to engage in further discussion.**

---

> > ### Comment · Reviewer_i1Ec · 2025-04-02
> >
> > Thanks to the authors for providing the rebuttal. I've read the author's response and comments from other reviewers. I have no further questions at this time. I will keep my original positive rating.

---

### Official Review · Reviewer_PBCW · 2025-03-12

**Overall Recommendation:** 3

**Summary:**

The authors explore the theoretical and empirical conditions under which neural networks could learn world models. To do so, they abstract the possible task that a model would need to learn as a set of Boolean functions which can be approximated using polynomials over binary inputs. The authors claim that this is a suitable abstraction since numerical representations in computers are represented with limited precision anyway and it allows to use simple complexity metrics. They then proceed to prove several theorems on how models would behave: will they learn shallow representations or deeper ones? The end by showing how their approach (fitting polynomials) performs against Transformers.


## update after rebuttal

I've raise my score to a week accept. I like the general direction of the paper and their responses (and my own independent analysis) have cleared some things up but I feel like it is really hard to read at times. The authors try to be very precise with their approach which I understand, but it makes following the paper harder than it should be, in my opinion.

**Claims And Evidence:**

There are several claims which do not convince me: that their representation abstraction (using polynomials over parity functions) is a reasonable proxy for the kind of representations that world models would operate on; that their tests correctly mimic what a world models would need to process.

The first is not at all clear to me because while it is true that computers operate with binary functions, human cognition does not. Sure brains operate with electrical signals, but the representations at the cognitive level are more abstract. Indeed the point of world models is that such abstractions emerge from this more basic substrate. It is not clear to me how such an equivalent process happens in this setting.

The second is related to this. If the input signals are so low-level and it is not clear if higher level abstractions emerge from them, that how do we know that complex functions as defined here mimic what a world model needs to learn?

**Essential References Not Discussed:**

No that I can recall.

**Experimental Designs Or Analyses:**

I checked them, they are fine.

**Methods And Evaluation Criteria:**

See above. The evaluations and methods make sense if we are evaluating the capability of networks to model functions using polynomials over parity functions, but I feel like equating these to world models is a stretch.

**Other Comments Or Suggestions:**

I would say that one way is to reframe this a problem of learning Boolean functions and not claim anything about world models.

**Other Strengths And Weaknesses:**

The paper tends to be well written, though it can get bogged down in very strict formality. On the other hand as a I have said it is not clear to me that this is a good abstraction of what world models must solve and the experimental evaluation is a bit limited.

**Questions For Authors:**

It is not clear when defining the problem setting how is this any different from the standard ICA, CRL, etc settings. And the authors clarification that "we do not assume that $p(z)$ is any structured distribution" is just vague. Can you provide a better definition?

In lines 264-268 you say "Indeed, modern pre-training objectives such as next-token prediction and contrastive learning can be interpreted as solving a large number of prediction tasks simultaneously (Radford et al., 2019; Arora et al., 2019)." But is this really true? Is it not stretching the definition of task over its implied meaning? For me, these are just ways to learn a distribution, which may or may not correspond to different tasks. What separates different tasks from different instances of a task in this framework?

**Relation To Broader Scientific Literature:**

As expressed above, I am not sure the findings are as relevant as the authors claim to the problem of learning world models.

**Theoretical Claims:**

No.

---

> ### Author Rebuttal · Authors · 2025-04-01
>
> Thank you for your insightful review. We provide point-by-point responses to your concerns in the following.
>
> **Definition of the world model:** Our abstraction of the world model is motivated by a line of research in the machine learning literature. In brief, the fundamental aspect of our world model definition is _"a model that learns a representation of high-level data-generating variables"_ (Definition 2.2). We emphasize that many works in the existing literature share a similar definition as ours (e.g., [1-4] and many others; see our responses to _Reviewer L9Q8_ for further elaboration on this point) and thus feel that our definition is well-motivated and captures the main spirit of learning world models _in the context of machine learning_.
>
>   - We did _not_ claim that our formulation precisely captures how the world model works in the brain. As you have mentioned in the review, the main point of our abstraction is to model how high-level representations/abstractions of data-generating variables can be learned from low-level inputs---note that this is also the main motivation of studying world models in the machine learning context [1]. On the other hand, we agree that how cognitive-level representations in the brain emerge might be a much harder question and for now, remains a mystery.
>
>   - Our world model definition does _not_ require Boolean functions: Note that Definition 2.2 (and the whole Section 2 that presents our formulation) does not depend on Boolean functions at all. Boolean functions are simply used as a _mathematical tool_ to characterize the impact of _low-complexity bias_  on identifying data-generating variables, which is otherwise difficult to analyze in the real domain (note that this tool has also been used by other machine learning works [5, 6]). That being said, we believe that this treatment still captures many interesting machine learning settings and applications, e.g., representation learning, out-of-distribution generalization, self-supervised learning, etc.
>
>   - Your review mentions that "If the input signals are so low-level and it is not clear if higher-level abstractions emerge from them...". We are not sure if we understood this correctly, but our formulation does require that high-level variables _can_ be obtained by inverting the data-generating function, which we believe is a minimal assumption to ensure that high-level variables are learnable from the inputs.
>
> **Detailed comparisons with ICA and CRL:** Thank you for your question. ICA assumes that different coordinates of the latent data-generating vector $\mathbf{z}$ are independent random variables, while CRL assumes that different coordinates of $\mathbf{z}$ are governed by a causal graph and aims to identify the causal relations between different coordinates. By contrast, Definition 2.1 does not impose any distributional constraints on the coordinates of $\mathbf{z}$, nor do our results rely on any of these assumptions. We will also add this discussion to the next version of our paper.
>
> **Multi-task learning:** The interpretation that pre-training objectives such as next-token prediction could be viewed as multi-task learning has been well-studied by prior work. Notably, the GPT-2 paper "Language models are unsupervised _multitask_ learners" [7] formulates next-token prediction as modeling the conditional distribution $p(output|input, task)$ with natural language as _task conditioning_. Task conditioning can also be implemented at an architectural level, i.e., using task-specific decoders as in our formulation.
>
> **Please let us know if you still have any questions or concerns. We are happy to engage in further discussion.**
>
> ---
>
> [1] Ha and Schmidhuber. World models. NeurIPS, 2018.
>
> [2] Li et al. Emergent world representations: Exploring a sequence model trained on a synthetic task. ICLR, 2023.
>
> [3] Richens and Everitt. Robust agents learn causal world models. ICLR, 2024.
>
> [4] Gurnee and Tegmark. Language models represent space and time. ICLR, 2024.
>
> [5] Barak et al. Hidden progress in deep learning: SGD learns parities near the computational limit. NeurIPS, 2022.
>
> [6] Emmanuel et al. Generalization on the unseen, logic reasoning and degree curriculum. ICML, 2023.
>
> [7] Radford et al. Language models are unsupervised multitask learners. 2019.

---

### Decision · Program_Chairs · 2025-05-01

**Decision:**

Accept (poster)

**Comment:**

This paper presents a novel theoretical framework for understanding when neural networks can learn "world models"—internal representations of latent data-generating variables. By combining a low-degree bias in Boolean transforms with multi-task learning, the authors prove identifiability results under mild assumptions. The work is well-motivated, connects to broader themes like out-of-distribution generalization and representation learning, and includes proof-of-concept experiments that demonstrate architectural implications.

While one reviewer questioned the abstraction’s connection to practical world models, the authors addressed this satisfactorily in the rebuttal. The reviewers generally appreciated the paper's theoretical depth and relevance. Overall, the paper provides foundational insights and is a valuable contribution to the theory of representation learning.